# Salus: Strategic Diagnostic Testing for Complex Diagnosis via Multi-Agent Reinforcement Learning

Shuohao Gao [* 1]  Xuanzhong Chen [* 1]  Lingxiao Luo [1]  Zilin Ding [1]  Rong Han [1]  Rui Jiang [2]  Ting Chen [1]

## Abstract

Diagnosing complex diseases is inherently a sequential and iterative medical investigation process, in which a clinician strategically requests multiple rounds of diagnostic tests to differentiate among similar diseases until reaching a definitive diagnosis. Although large language models show great potential as clinical assistants, they often struggle to navigate this complex interactive process, suffering from premature diagnostic closure. Furthermore, optimizing LLMs for such multi-round environments is frequently hindered by the challenge of reward sparsity and hacking. In this paper, we introduce **CompDiag-Bench**, a benchmark that formalizes diagnosis as a sequential decision-making process where a clinician must strategically request diagnostic tests from a dynamic environment in order to reach a definitive diagnosis. To address this task, we propose Salus, a multi-agent framework that decouples diagnostic reasoning into three specialized functional roles: a Differential Reasoner, a Strategic Controller, and a Workup Proposer. Salus is optimized via multi-agent reinforcement learning employing structured rewards to calibrate strategic diagnostic behavior. Specifically, we leverage an LLM-as-a-Judge reward mechanism to provide dense, semantically-grounded feedback, designed to penalize premature closure and incentivize accurate differential diagnoses. Experimental results show that our model, Salus-7B, attains state-of-the-art Top-1 accuracy of 83.64% on complex cases, outperforming DeepSeek-V3.2 (71.38%) and achieving performance on par with GPT-5.2 (80.30%).

---
[*]Equal contribution  [1]Department of Computer Science and Technology, Tsinghua University, Beijing, China [2]Department of Automation, Tsinghua University, Beijing, China. Correspondence to: Ting Chen <tingchen@tsinghua.edu.cn>.

*Proceedings of the $43^{rd}$ International Conference on Machine Learning*, Seoul, South Korea. PMLR 306, 2026. Copyright 2026 by the author(s).

## 1. Introduction

In clinical practice, complex diagnosis is a dynamic and sequential process rather than a static question-answering task. Neither is it to predict diseases based on the Electronic Health Records (EHRs). The diagnostic process starts from an initial clinical context about the history of present illness, physical examination findings, and preliminary results, and then evolves through multiple rounds of acquiring and interpreting additional investigations (Pelaccia et al., 2011). Within this paradigm, a clinician must strategically request diagnostic testings to resolve clinical uncertainty and refine hypotheses until a definitive diagnosis is reached. As illustrated in Figure 1, consider a patient presenting with dyspnea and palpitations. A superficial analysis might suggest a stress-induced panic attack, leading to *premature closure* and a misdiagnosis of Anxiety Disorder (Path 1). Conversely, a competent clinician must recognize subtle danger signals (e.g., prolonged immobility) and resolve uncertainty by requesting targeted investigations such as D-Dimer to uncover the life-threatening root cause (Acute Pulmonary Embolism in Path 2). Such interactive task necessitates three critical capabilities: synthesizing differential diagnoses from incomplete and evolving clinical records; proposing targeted diagnostic tests; and determining the optimal termination moment to balance diagnostic accuracy against the risks of premature closure or excessive testing.

Recently, Large Language Models (LLMs) have demonstrated remarkable proficiency in the medical domain, attaining high performance on established benchmarks such as MedQA (Jin et al., 2021) and various medical licensing examinations (Nori et al., 2023; Cabral et al., 2024; Goh et al., 2024; McDuff et al., 2025). These advancements suggest that LLMs have acquired extensive medical knowledge and are capable of complex clinical synthesis when presented with comprehensive case data. However, such success in single-round settings may not adequately reflect the clinical utility of LLMs in practice (Tu et al., 2025; Nori et al., 2025). Real-world medicine is a dynamic and fundamentally more challenging process, where the primary difficulty resides not merely in the final prediction, but in the strategic navigation of clinical uncertainty through sequential interaction.

To bridge this gap, we introduce **CompDiag-Bench**, a

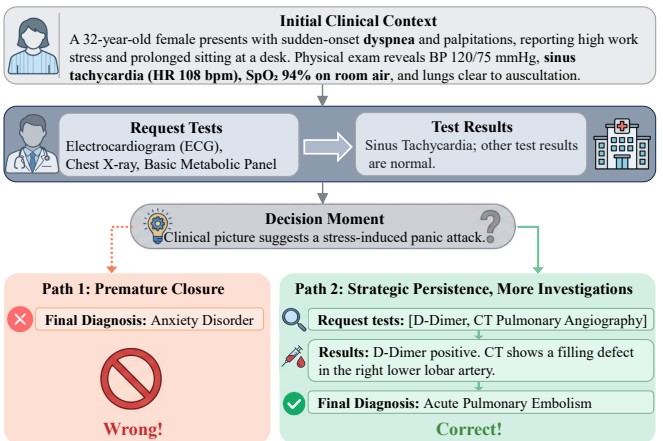

*Figure 1.* An example of strategic diagnostic testing. Faced with vague initial symptoms (dyspnea and tachycardia), the clinician must make a critical decision: conclude the diagnosis or request further tests. *Path 1* illustrates a common failure mode where the clinician settles for a plausible but incorrect diagnosis based on surface-level heuristics, suffering from premature closure. *Path 2* shows the ideal workflow, where the clinician identifies residual uncertainty and insufficient evidence for current Top-1 diagnosis, requests targeted diagnostic tests including Computed Tomography (CT), and finally correctly identifies the high-risk pathology.

benchmark that formalizes complex diagnosis as a sequential interaction between an artificial intelligence system (AI clinician) and an automated *Diagnostic Environment*. Utilizing 752 curated high-quality records, the environment provides the results of multiple test items from the AI clinician, based on the original clinical records. Our evaluation reveals that current open-source LLMs, including specialized medical models, struggle to navigate this complex environment. As shown in Figure 2, prominent models such as DeepSeek-V3.2 and Baichuan-M2-32B achieve Top-1 accuracies of 71.4% and 48.7%, respectively, frequently suffering from strategic errors or premature diagnostic closure. Furthermore, optimizing LLMs for such interactive tasks through reinforcement learning (RL) faces dual hurdles: 1) *reward sparsity* when the rewards solely rely on final diagnostic accuracy (Zou et al., 2024), and 2) *reward hacking* when providing intermediate signals for specific tests (Qiu et al., 2025a), which often leads the model to blindly mimic historical records rather than reasoning from evidence.

To address these challenges, we propose Salus, a multi-agent system that mirrors the human cognitive paradigm by decoupling clinical reasoning into three specialized agents: a *Differential Reasoner* for clinical synthesis, a *Strategic Controller* for termination logic, and a *Workup Proposer* for requesting targeted tests. This modular architecture allows us to deconstruct the multi-round interaction into a series of single-round decision instances, thereby transforming the traditional long-horizon RL problem (He & Chen, 2022;

He et al., 2022) into a step-wise optimization task, which alleviates the credit assignment problem and mitigates both reward sparsity and hacking. Based on this framework, Salus is optimized through multi-agent RL, initialized via a reasoning-aware supervised fine-tuning (SFT). Specifically, we leverage an LLM-as-a-Judge mechanism (Zheng et al., 2023) to implement a tiered reward structure, delivering *dense, semantically-grounded* feedback to penalize premature diagnostic closure and prioritize accurate differential diagnoses. By applying our optimization framework, we improve the Top-1 accuracy of a 7B backbone from 27.9% to 83.6%, surpassing the 71.4% achieved by DeepSeek-V3.2 and reaching a level comparable to GPT-5.2.

Our main contributions are as follows:

- We introduce **CompDiag-Bench**, an interactive framework that formalizes diagnosis as a sequential decision-making process, which utilizes 752 curated records containing both complex and routine cases to simulate realistic and multi-round diagnostic workflows.

- We propose Salus, based on a multi-agent architecture that decouples clinical reasoning from strategic control. We optimize it through an RL training with LLM-as-a-Judge rewards, providing dense feedback to mitigate premature closure.

- Our Salus achieves state-of-the-art (SOTA) performance, demonstrating that a 7B model can outperform prominent open-source models such as DeepSeek-V3.2 and match the performance of GPT-5.2 and Gemini 3 Flash on high-complexity cases.

## 2. The Complex Diagnosis Benchmark

In this section, we introduce **CompDiag-Bench** (Complex Diagnosis Benchmark), a framework designed to benchmark the capability of an *AI clinician* to navigate complex clinical scenarios. The framework formalizes the diagnostic process as a sequential interaction, requiring the AI clinician to strategically request diagnostic testing to resolve clinical uncertainty and finally reach the accurate diagnosis.

**Task Description.** We model the diagnostic task as a sequential decision-making process aimed at maximizing diagnostic accuracy within acceptable testing efficiency. Unlike simplified medical question-answering (QA) tasks (Arora et al., 2025; Jin et al., 2019) or diagnosis-only tasks (Zhu et al., 2025; Chen et al., 2024b), our framework emphasizes the *interactive* nature of clinical medicine. At the beginning of each session, the AI clinician is provided with an **initial clinical context** comprising a comprehensive patient profile, including the chief complaint, history of present illness, physical examination findings, and prior test results

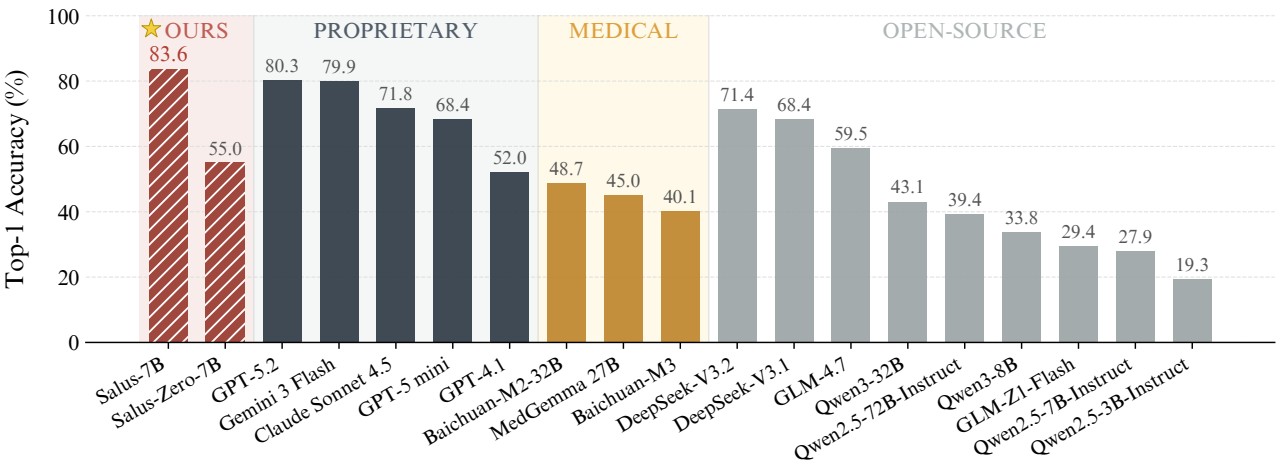

*Figure 2.* Top-1 diagnostic accuracy on 269 high-complexity cases from CompDiag-Bench. `Salus-7B` achieves SOTA performance, outperforming leading open-source models (e.g., DeepSeek-V3.2) and matching the performance of GPT-5.2 in complex clinical synthesis.

as shown in Figure 3(b). These elements establish the foundational context, simulating clinical scenarios ranging from standard intake to complex medical referrals. The core interaction evolves through **sequential diagnostic testing**, where the AI clinician iteratively requests investigations such as specialized imaging, laboratory panels, or invasive procedures. To mirror realistic clinical workflows, the AI clinician is permitted to request *multiple diagnostic test items per round*, which districts from existing interactive benchmarks that often restrict clinicians to request single examination (Qiu et al., 2025a; Schmidgall et al., 2024). This interactive process continues until the AI clinician submits a final diagnostic conclusion.

**Data Curation.** We curated a comprehensive patient record dataset to support both the training of AI clinicians and benchmarking of the **CompDiag-Bench** framework. As shown in Figure 3(a), a record extraction pipeline, utilizing DeepSeek-V3.1, standardizes raw clinical text into organized records containing four core components: *medical history*, *physical examination findings*, *diagnostic test results*, and the *gold-standard final diagnosis*. Specifically, the medical history and physical examination findings constitute the initial clinical context as mentioned above.

The curated dataset is partitioned into a training set of 8,645 cases and a multi-source test set of 752 cases. Specifically, all cases in the training set are restricted to publications prior to January 1, 2024, to prevent temporal data leakage. The test set comprises 269 patient records from prestigious journals (e.g., the New England Journal of Medicine), which are predominantly high-complexity, along with 483 relatively routine cases. This high complexity was verified by a systematic 5-level difficulty taxonomy (Appendix B.2). To ensure diagnostic validity, each case in the test set was

verified by both LLMs (DeepSeek-V3.1 and GPT-5 mini) as well as human experts; only cases achieving consensus on a definitive diagnosis supported by sufficient evidence were retained, ensuring that the gold-standard diagnoses are robustly justified by the documented findings.

**Interactive Diagnostic Environment.** To facilitate multi-round interaction, we develop the *Diagnostic Environment* by prompting an LLM to act as a proxy for the patient and the hospital's diagnostic infrastructure, delivering test results in response to clinician requests. The environment's responses are grounded in an original, high-fidelity patient record that contains a gold-standard diagnosis. In instances where the clinician requests a test not explicitly documented in the patient record, the environment generates a synthetic finding constrained to be *pathophysiologically consistent* with the underlying disease and the patient's specific clinical state (Nori et al., 2025). To ensure rigorous evaluation, the environment is strictly prohibited from leaking diagnostic impressions, compelling the AI clinician to derive diagnoses solely through the interpretation of objective evidence. In a human audit of 100 randomly sampled environment responses, only 7 cases contain potentially diagnostic terms, all of which are pathology or biopsy reports (e.g., reporting *malignant cells* for a cancer patient), reflecting realistic clinical feedback rather than direct diagnosis leakage.

**Evaluation Metrics and Adjudication.** As illustrated in Figure 3(b), we evaluate performance using two primary metrics: *1) Diagnostic Accuracy:* Assesses the AI clinician's ability to reach the correct conclusion. By prompting an LLM as a *Judge model*, the finally predicted Top-3 diagnoses are evaluated against the gold-standard diagnosis. *2) Information Sufficiency:* Quantifies the AI clinician's effectiveness in eliciting critical clinical evidence through

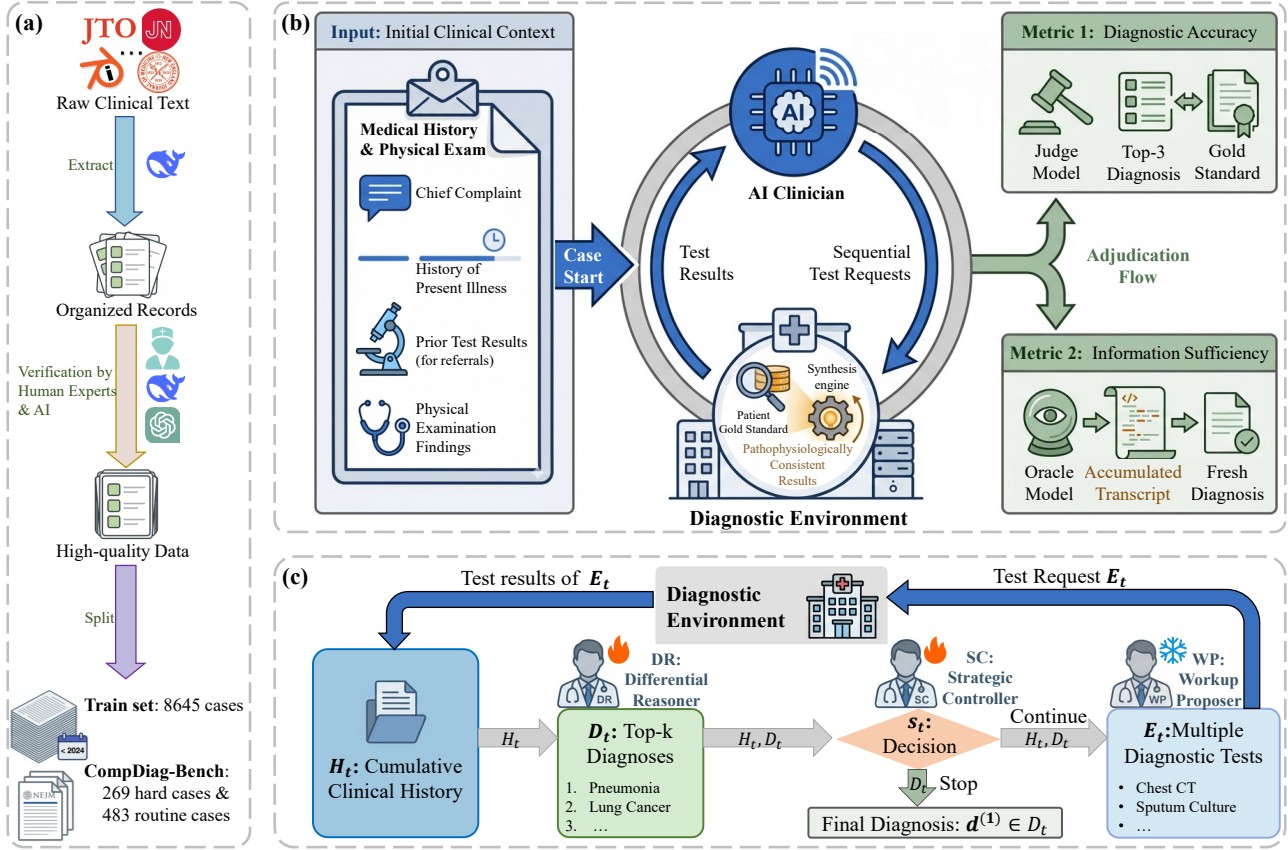

Figure 3. **Overview of the CompDiag-Bench and Salus.** (a) Data Curation: Extraction of organized records from multi-source clinical corpora. The benchmark includes 269 high-complexity cases and 483 routine records, which were verified by both LLMs and human experts. (b) Sequential Diagnostic Workflow: The AI clinician iteratively interacts with the Diagnostic Environment, starting from the initial clinical context; performance is adjudicated using Diagnostic Accuracy and Information Sufficiency, which quantify the completeness of the gathered clinical evidence. (c) Workflow of Salus: At each step $t$, the DR generates differential diagnoses $D_t$, followed by the SC determining the signal $s_t$. If $s_t$ = Continue, the WP proposes tests $E_t$, and the history $H_{t+1}$ is updated with results provided by Diagnostic Environment; otherwise, the top hypothesis $d^{(1)}$ is submitted as the final diagnosis.

multi-round interaction. The accumulated transcript, consisting of the initial context and the results of requested tests (excluding the predicted diagnoses), is submitted to an independent *Oracle model* for a fresh diagnosis. The accuracy of this fresh diagnosis serves as a proxy for the completeness of information gathered during the interaction.

To ensure diagnostic validity, diagnostic correctness is evaluated using a four-level clinician-authored rubric as detailed in Table 1, which assesses the alignment of the core disease entity and its potential impact on clinical management. A diagnosis is considered *correct* only if categorized as *Completely Correct* or *Clinically Acceptable*, ensuring that a correct diagnosis from the AI clinician is functionally equivalent to the gold-standard diagnosis in the record and would lead to a consistent clinical management plan. Furthermore, to validate this LLM-based automated adjudication protocol, we conduct a blind study in which three board-certified

physicians from respiratory medicine, gastroenterology, and cardiology evaluate 240 diagnoses sampled from the test trajectories (in Section 4.1), according to the same four-level rubric. The LLM judge (using DeepSeek-V3.1) shows substantial agreement with human physician annotations under the four-level rubric (Cohen's $\kappa = 0.64$) and almost perfect agreement after binarizing labels into *correct* versus *not correct* (Cohen's $\kappa = 0.81$), confirming that the Diagnostic Accuracy metric with 4-level rubric is highly aligned with expert clinical judgment.

Beyond these core evaluation axes, we report two auxiliary efficiency metrics: number of testing rounds and total number of requested test items. However, we note that efficiency is secondary in high-stakes clinical settings. Our evaluation prioritizes diagnostic accuracy and information sufficiency within reasonable efficiency, as premature termination to minimize costs might lead to catastrophic misdiagnosis.

*Table 1.* A four-level rubric for diagnosis evaluation based on core disease matching and clinical management impact. Standard medical synonyms (e.g., Hives vs. Urticaria) are treated as equivalent.

| LABEL | DEFINITION |
|---|---|
| **Completely Correct** | Matches all core diagnoses; clinically identical to gold-standard diagnosis. |
| **Clinically Acceptable** | Core direction is correct with minor precision flaws; differences do not affect subsequent treatment plans. |
| **Partially Correct** | Clinically relevant but contains critical omissions or vagueness that significantly impact management. |
| **Incorrect** | Fundamentally incorrect, entirely deviated from gold-standard diagnosis. |

# 3. `Salus`: Implementation and Optimization

We present `Salus`, a modular AI clinician designed to navigate complex diagnostic process. `Salus` employs a multi-agent architecture to decouple clinical reasoning from strategic control, and is optimized through a two-stage paradigm: reasoning-aware SFT followed by multi-agent RL.

## 3.1. Multi-Agent Framework

To formalize the intricate cognitive processes required for complex clinical reasoning, we propose a multi-agent framework that decomposes the *AI clinician* into three specialized functional agents: the *Differential Reasoner (DR)*, the *Strategic Controller (SC)*, and the *Workup Proposer (WP)*. This architecture is intentionally designed to mirror the clinical reasoning paradigm of human clinicians: (1) the DR handles the high-stakes task of clinical synthesis to generate a top-$k$ differential diagnoses ($k$ is a hyper-parameter); (2) the SC serves as a meta-cognitive governor to determine whether the current evidence is sufficient, acting as the primary safeguard against premature closure and misdiagnosis; (3) the WP translates diagnostic uncertainty into a targeted list of diagnostic investigations.

Formally, let $H_t$ denote the cumulative clinical history at interaction step $t$, encompassing the initial patient profile and the results of all preceding investigations. The diagnostic workflow is illustrated in Figure 3(c). At each iteration, the DR processes $H_t$ to generate a ranked list of differential diagnoses $D_t = \{d^{(1)}, d^{(2)}, \ldots, d^{(k)}\}$. Subsequently, the SC evaluates the pair $(H_t, D_t)$ to produce a binary decision $s_t \in \{\text{Stop, Continue}\}$. If the investigation continues, the WP requests a set of diagnostic tests $E_t$, which are then executed within the *Diagnostic Environment*. The environment returns testing results that are incorporated to update the clinical history to $H_{t+1}$. Upon a *Stop* signal or reaching the maximum round limit $T_{max}$, the primary hypothesis, $d^{(1)}$, is emitted as the final diagnosis.

**Rationale and Advantages.** Beyond aligning with clinical practice and providing modular controllability, this tripartite decomposition is primarily designed to overcome the fundamental training bottlenecks of RL in complex diagnostic trajectories. In a monolithic (single-agent) architecture, diagnostic agents typically suffer from *sparse rewards*, as definitive feedback is often only available upon final diagnosis. Conversely, attempting to provide intermediate rewards for specific test requests, such as rewarding the model for matching the exact examinations documented in the original record (e.g., the Complete Blood Count), frequently induces *reward hacking*. In such cases, the agent may overfit to the historical record's specific testing path while ignoring clinically equivalent or superior alternatives (Qiu et al., 2025a). By decoupling the process, `Salus` enables a *dense and step-wise reward structure*, as detailed in Section 3.2.2.

## 3.2. Optimization of `Salus`

### 3.2.1. REASONING-AWARE COLD START.

To imbue the model with foundational medical reasoning capabilities and role-specific expertise, we initiate the training process with a reasoning-aware cold start. This phase is designed to bootstrap the AI clinician's ability to perform complex clinical synthesis and manage the diagnostic workflow by distilling expertise from a high-capacity teacher model. Specifically, we employ a teacher model (e.g., DeepSeek-V3.1) to generate high-quality clinical trajectories by performing full-trajectory rollouts as illustrated in Figure 3(c). During each interaction, the teacher model sequentially assumes the roles of DR, SC, and WP by switching between task-specific prompts (or instructions $\mathcal{I}_i$) where $i \in \{\text{DR, SC, WP}\}$. For each interaction step $t$, we define the input $X_i$ and the corresponding response $Y_i$ as:

$$
\begin{aligned}
X_{\text{DR}} &= (\mathcal{I}_{\text{DR}}, H_t), & Y_{\text{DR}} &= \{\text{CoT}_t^{\text{DR}}, D_t\} \\
X_{\text{SC}} &= (\mathcal{I}_{\text{SC}}, H_t, D_t), & Y_{\text{SC}} &= \{\text{CoT}_t^{\text{SC}}, s_t\} \quad (1) \\
X_{\text{WP}} &= (\mathcal{I}_{\text{WP}}, H_t, D_t), & Y_{\text{WP}} &= \{\text{CoT}_t^{\text{WP}}, E_t\}
\end{aligned}
$$

where $\text{CoT}_t^i$ denotes the long-form Chain-of-Thought reasoning generated within `<think>` tags, and $D_t, s_t, E_t$ represent the functional outputs for differential diagnosis, termination control, and diagnostic tests, respectively.

To ensure strategic rigor, we implement a *Logic-Consistency Filter* on the SC's training data. Specifically, we discard any input-response pair $(X_{\text{SC}}, Y_{\text{SC}})$ where the teacher model issues a premature stop signal ($s_t = \text{Stop}$) while the primary diagnosis $d^{(1)}$ remains incorrect. This ensures the model learns to prioritize evidence sufficiency over premature closure. Given that all roles are executed by a unified backbone model $\theta$, the SFT objective is to maximize the log-likelihood

across the aggregated role-specific datasets $\mathcal{S}_i$:

$$\sum_{i \in \{\text{DR, SC, WP}\}} \mathbb{E}_{(X_i, Y_i) \sim \mathcal{S}_i} \log P_\theta(Y_i \mid X_i)$$

Note that all three agents are optimized during SFT stage. This multi-task initialization internalizes the collaborative logic required for the tripartite agent architecture.

### 3.2.2. MULTI-AGENT REINFORCEMENT LEARNING

Building upon the SFT-initialized parameters, we employ RL to further refine the agents' decision-making. In this stage, we focus on the joint optimization of the *Differential Reasoner (DR)* and the *Strategic Controller (SC)*, while the *Workup Proposer (WP)* is not optimized through RL.

**LLM-as-a-Judge Reward Modeling.** Evaluating clinical diagnoses exceeds the capabilities of string-matching or rule-based heuristics due to the high linguistic variability and synonymy in medical terminology (e.g., "STEMI" vs. "Myocardial Infarction"). We therefore employ an *LLM-as-a-Judge* to adjudicate the functional and semantic alignment between agent predictions and gold-standard diagnoses. By mapping qualitative clinical reasoning onto the four-level rubric in Table 1, this framework generates the semantically-grounded reward signals required for effective RL.

**1) Differential Reasoner Reward ($\mathcal{R}_{\text{DR}}$):** The DR is incentivized to prioritize the most clinically sound hypothesis. Crucially, in the early stages of a clinical encounter, the gold-standard diagnosis may not yet be the statistically most probable outcome (Top-1 diagnosis) given the initial sparse evidence. Expecting the model to predict the gold-standard diagnosis as Top-1 diagnosis prematurely would encourage "hallucinated leaps" unsupported by evidence. To account for this inherent clinical uncertainty and the conditional probability of disease given evolving evidence, we employ a tiered reward structure that encourages the DR to maintain the gold-standard diagnosis within the Top-3 candidates.

Formally, let $D_t = \{d^{(1)}, d^{(2)}, \ldots, d^{(k)}\}$ be the ranked list of $k$ differential diagnoses generated by the DR at step $t$ (which derives from $Y_{\text{DR}}$), the judge assigns a reward based on the highest-ranking correct diagnosis by comparing the predicted diagnosis $d^{(j)} \in D_t$ with the gold-standard diagnosis documented in the patient record:

$$\mathcal{R}_{\text{DR}}(Y_{\text{DR}}) = \begin{cases} 4, \text{if } d^{(1)} \text{ is Completely Correct} \\ 3, \text{else if } d^{(1)} \text{ is Clinically Acceptable} \\ 2, \text{else if } d^{(2)} \text{ is Correct} \\ 1, \text{else if } d^{(3)} \text{ is Correct} \\ 0, \text{otherwise} \end{cases}$$

This tiered structure alleviates the *sparse reward* problem and incentivizes the model to iteratively refine its differential list as more evidence is gathered.

**2) Strategic Controller Reward ($\mathcal{R}_{\text{SC}}$):** The SC reward is engineered to penalize premature closure while enforcing confidence when evidence is sufficient. Let $s_t$ be the output signal (which derives from $Y_{\text{SC}}$), the optimization objective considers three distinct clinical states:

- **Case 1 (Correct Top-1 with Sufficient Information):** If $d^{(1)} \in D_t$ is correct and the current history $H_t$ is adjudicated by a judge model as sufficient to support $d^{(1)} \in D_t$, the SC is rewarded only for terminating the session ($s_t =$ Stop). This prevents unnecessary over-testing.

$$\mathcal{R}_{\text{SC}}(Y_{\text{SC}}) = \begin{cases} 1, \text{if } s_t = \text{Stop} \\ 0, \text{otherwise} \end{cases}$$

- **Case 2 (Insufficient Information but Correct Top-1):** If $d^{(1)} \in D_t$ is correct with insufficient supporting evidence, a positive reward is given for Stop *only* if the diagnosis is *Completely Correct*. If the diagnosis is merely *Clinically Acceptable*, the reward is neutral (i.e., 0) for both Stop and Continue. This reflects the clinical reality where an *Acceptable* diagnosis might suffice for management but still lacks the precision of a *Complete* diagnosis, thereby neither strictly mandating nor discouraging further workup.

$$\mathcal{R}_{\text{SC}}(Y_{\text{SC}}) = \begin{cases} 1, \text{if } s_t = \text{Stop} \wedge d^{(1)} \text{ is Completely Correct} \\ 0, \text{otherwise} \end{cases}$$

- **Case 3 (Incorrect Diagnosis):** If the primary hypothesis $d^{(1)} \in D_t$ is incorrect, the SC is penalized for stopping, compelling the clinician to seek further evidence.

$$\mathcal{R}_{\text{SC}}(Y_{\text{SC}}) = \begin{cases} 1, \text{if } s_t = \text{Continue} \\ 0, \text{otherwise} \end{cases}$$

This reward structure formalizes a risk-averse termination policy by inducing a clear preference hierarchy: *Completely Correct* $\succ$ {*Clinically Acceptable, Continued Investigation*} $\succ$ *Misdiagnosis*, mitigating the risk of premature closure. Specifically, instead of forcing the AI physician to make a final diagnosis (which may be correct or not), we additionally offer it another option: the AI physician can reject making a final diagnosis and continue investigating. This allows LLMs to avoid hallucination and premature closure.

**3) Workup Proposer:** We note that the WP is not optimized during RL phase. First, it is challenging to directly obtain the ground truth of the WP's actions from the original patient record. The gold-standard final diagnosis cannot provide direct supervision, nor can it be easily used to design rewards. Second, the action space of the WP is complex and involves multiple evaluation axes, including the cost of diagnostic tests, the ability to differentiate among candidate

diagnoses $D_t$, and the invasiveness to the patient, etc. Therefore, optimizing the WP via numerical reward design during the RL phase is very difficult and often leads the model to reward hacking.

**Optimization Objective.** We jointly optimize the policy $\pi_\theta$ for the DR and SC components by deconstructing the sequential diagnostic process into independent decision-making instances. While full-trajectory RL is theoretically ideal, the significant inference latency and resource overhead of an LLM-driven *Diagnostic Environment* make synchronous multi-round interactions computationally prohibitive. To ensure tractability, we sample clinical states $X_i$ directly from the SFT datasets $\mathcal{S}_{DR} \cup \mathcal{S}_{SC}$. Crucially, unlike offline RL which learns from fixed historical transitions, our approach performs online rollouts for each sampled state $X_i$. Specifically, given these state-response pairs, the policy $\pi_\theta$ is optimized to maximize the expected reward while remaining regularized against the SFT-initialized reference model $\pi_{ref}$ via a Kullback-Leibler (KL) divergence penalty. The final objective is to maximize the following formulation:

$$\sum_{i \in \{DR, SC\}} \mathbb{E}_{X_i \sim \mathcal{S}_i} \left[ \mathbb{E}_{Y_i \sim \pi_\theta(\cdot|X_i)}[\mathcal{R}_i(Y_i)] - \beta \mathbb{D}_{KL}(\pi_\theta \| \pi_{ref}) \right]$$

where the coefficient $\beta$ controls the strength of KL penalty. We note that only the DR agent and the SC agent are optimized during reinforcement learning stage.

**Advantage Analysis.** This optimization paradigm yields dual benefits in efficiency and stability: 1) It circumvents the prohibitive computational overhead and interaction latency inherent in real-time, multi-round environment rollouts. 2) By decomposing long-horizon trajectories, it provides dense reward signals that effectively alleviate credit assignment challenges and mitigate the high gradient variance typical of multi-round RL.

# 4. Experiments

**Setup.** We instantiate Salus using the Qwen2.5-7B-Instruct backbone. Optimization follows the two-stage paradigm detailed in Section 3: a reasoning-aware SFT, followed by RL via Group Relative Policy Optimization (GRPO) (Shao et al., 2024) with a group size of $G = 8$ rollouts per state. DeepSeek-V3.2 serves as the automated adjudicator for the *LLM-as-a-Judge* reward mechanism (Section 3.2.2). We denote the SFT-only baseline as Salus-Zero-7B and the final RL-optimized model as Salus-7B. To instantiate the interactive framework, DeepSeek-V3.1 is employed as the *Diagnostic Environment*, the *Judge*, and the *Oracle* model (Section 2). During evaluation, the AI clinician is restricted to request at most $T_{max} = 8$ rounds of tests, and the DR agent is configured to provide a top-$k$ differential list with $k = 5$. The interaction history, presented in a multi-round dialog format,

serves as the cumulative clinical history $H_t$, thereby preventing potential information loss. We report performance via Top-3 accuracy and Information Sufficiency (Inf. Suff.), while characterizing interaction efficiency using the average number of test rounds (#Rounds) and the total count of requested test items (#Tests).

**Baseline Models.** We benchmark Salus against a diverse suite of state-of-the-art (SOTA) LLMs, including both proprietary models (e.g., GPT-5.2, Gemini 3 Flash) and prominent general-purpose open-source LLMs and medical-specific LLMs. To ensure a rigorous and controlled comparison, all baseline models are integrated into the same tripartite multi-agent architecture described in Section 3.1 via task-specific prompting. A complete list of baseline model descriptions is provided in Appendix C.

The source code and dataset are available at https://github.com/ShuohaoGao/ICML-Salus.

## 4.1. Main Results

**Results on High-Complexity Cases.** We evaluate the strategic diagnostic capabilities of Salus and baseline LLMs on the 269 high-complexity patient records sourced from prestigious medical journals. The results are summarized in Figure 2 and Table 2. Salus-7B attains the SOTA Top-1 accuracy of 83.64%, outperforming all general-purpose open-source, and specialized medical LLMs. Specifically, Salus-7B surpasses DeepSeek-V3.2 by 12.26% (absolute) and demonstrates performance competitive with frontier proprietary models such as GPT-5.2 and Gemini 3 Flash. Furthermore, the optimization of RL yields a significant performance leap; Salus-7B improves upon its SFT counterpart, Salus-Zero-7B, by 28.62% in Top-1 accuracy. Notably, even Salus-Zero-7B (55.02%) exhibits superior diagnostic performance compared to substantially larger models like Qwen3-32B and Baichuan-M2-32B, validating the efficacy of our reasoning-aware cold-start strategy.

**Generalization to Routine Clinical Cases.** To evaluate the robustness and generalization of Salus across varying clinical complexities, we benchmarked the models on 483 routine cases that typically exhibit lower diagnostic difficulty compared to the journal-sourced high-complexity cases. As summarized in Table 3, the performance trends largely mirror those observed in the high-complexity evaluation, and Salus-7B maintains its SOTA performance with 91.10% Top-1 accuracy. These results suggest the effectiveness of our optimization methods in Section 3.

## 4.2. Ablation Study

We conduct ablation studies on 269 high-complexity cases to evaluate the factors contributing to Salus's performance.

**Analysis of Reasoning-aware Cold-start.** As shown in

*Table 2.* Performance comparison on 269 high-complexity cases. **Top-1/2/3 Accuracy** assesses diagnostic precision; **Inf. Suff.** measures the completeness of evidence gathered. Interaction efficiency is shown via testing rounds (#Rounds) and total test items (#Tests). Best results are **bolded**; second-best results are underlined; 95% confidence intervals (95% CI) are reported for primary metrics.

| LLM | Top-1 (95% CI) | Top-2 (95% CI) | Top-3 (95% CI) | Inf. Suff. ↑ (95% CI) | #Rounds | #Tests |
|---|---|---|---|---|---|---|
| *Proprietary LLMs* | | | | | | |
| GPT-5.2 | 80.30 (75.1–84.6) | 85.50 (80.8–89.2) | 86.25 (81.6–89.9) | 71.00 (65.3–76.1) | 1.75 | 27.61 |
| Gemini 3 Flash | 79.93 (74.7–84.3) | 84.39 (79.6–88.2) | 86.62 (82.0–90.2) | 72.12 (66.5–77.1) | 0.85 | 8.26 |
| Claude Sonnet 4.5 | 71.75 (66.1–76.8) | 76.95 (71.6–81.6) | 79.93 (74.7–84.3) | 70.63 (64.9–75.8) | 1.19 | 14.75 |
| GPT-5 mini | 68.40 (62.6–73.7) | 75.09 (69.6–79.9) | 76.95 (71.6–81.6) | 63.57 (57.7–69.1) | 1.06 | 14.82 |
| GPT-4.1 | 52.05 (46.1–57.9) | 59.85 (53.9–65.5) | 62.08 (56.2–67.7) | 55.76 (49.8–61.6) | 0.89 | 5.82 |
| *Open-source LLMs* | | | | | | |
| DeepSeek-V3.2 | 71.38 (65.7–76.4) | 77.32 (72.0–81.9) | 78.44 (73.1–82.9) | 66.91 (61.1–72.3) | 1.88 | 11.95 |
| DeepSeek-V3.1 | 68.40 (62.6–73.7) | 73.23 (67.6–78.2) | 77.32 (72.0–81.9) | 68.40 (62.6–73.7) | 2.55 | 10.56 |
| GLM-4.7 | 59.48 (53.5–65.2) | 65.06 (59.2–70.5) | 68.77 (63.0–74.0) | 60.60 (54.6–66.2) | 0.87 | 6.89 |
| Qwen3-32B | 43.12 (37.3–49.1) | 49.81 (43.9–55.7) | 53.53 (47.6–59.4) | 52.05 (46.1–57.9) | 1.11 | 7.00 |
| Qwen2.5-72B-Instruct | 39.40 (33.8–45.4) | 46.84 (41.0–52.8) | 49.44 (43.5–55.4) | 54.65 (48.7–60.5) | 1.07 | 7.45 |
| Qwen3-8B | 33.83 (28.4–39.7) | 40.15 (34.5–46.1) | 43.87 (38.1–49.8) | 54.27 (48.3–60.1) | 1.51 | 9.23 |
| GLM-Z1-Flash | 29.37 (24.2–35.1) | 37.18 (31.6–43.1) | 43.12 (37.3–49.1) | 54.65 (48.7–60.5) | 2.38 | 14.87 |
| Qwen2.5-7B-Instruct | 27.88 (22.9–33.5) | 35.69 (30.2–41.6) | 40.15 (34.5–46.1) | 59.11 (53.1–64.8) | 2.41 | 18.08 |
| Qwen2.5-3B-Instruct | 19.33 (15.1–24.5) | 24.91 (20.1–30.4) | 26.76 (21.8–32.4) | 57.25 (51.3–63.0) | 2.37 | 14.27 |
| *Medical LLMs* | | | | | | |
| Baichuan-M2-32B | 48.70 (42.8–54.6) | 53.53 (47.6–59.4) | 56.13 (50.2–61.9) | 56.51 (50.5–62.3) | 3.10 | 25.98 |
| MedGemma 27B | 44.98 (39.1–51.0) | 52.79 (46.8–58.7) | 57.25 (51.3–63.0) | 58.74 (52.8–64.5) | 1.38 | 12.19 |
| Baichuan-M3 | 40.15 (34.5–46.1) | 43.49 (37.7–49.5) | 46.10 (40.2–52.1) | 53.53 (47.6–59.4) | 1.31 | 6.28 |
| Salus-Zero-7B | 55.02 (49.0–60.9) | 59.11 (53.1–64.8) | 62.08 (56.2–67.7) | 65.80 (59.9–71.2) | 2.54 | 12.51 |
| Salus-7B | **83.64** (78.8–87.6) | **86.62** (82.0–90.2) | **88.85** (84.5–92.1) | **72.86** (67.3–77.8) | 2.91 | 19.98 |

*Table 3.* Diagnostic performance on 483 routine cases. We report Top-1 Accuracy (Top-1) and Information Sufficiency (Inf. Suff.). Full results are provided in Table 11 in Appendix A.

| LLM | Top-1 (95% CI) | Inf. Suff. ↑ (95% CI) |
|---|---|---|
| *Proprietary LLMs* | | |
| Gemini 3 Flash | 88.20 (83.7–91.4) | **84.06** (79.2–87.9) |
| GPT-5.2 | 86.34 (81.6–89.9) | 80.75 (75.5–84.9) |
| GPT-5 mini | 80.95 (75.9–85.3) | 83.85 (79.2–87.9) |
| *Open-source LLMs* | | |
| DeepSeek-V3.2 | 82.19 (77.1–86.3) | 82.82 (77.9–86.9) |
| DeepSeek-V3.1 | 81.57 (76.3–85.6) | 81.57 (76.3–85.6) |
| Qwen3-32B | 76.19 (70.8–80.9) | 76.19 (70.8–80.9) |
| Qwen3-8B | 66.87 (61.1–72.3) | 76.40 (71.2–81.2) |
| Qwen2.5-7B-Instruct | 59.63 (53.5–65.2) | 80.95 (75.9–85.3) |
| *Medical LLMs* | | |
| MedGemma 27B | 75.16 (69.6–79.9) | 80.95 (75.9–85.3) |
| Baichuan-M2-32B | 58.18 (52.4–64.1) | 62.73 (56.9–68.4) |
| Salus-Zero-7B | 78.67 (73.5–83.3) | 83.23 (78.3–87.3) |
| Salus-7B | **91.10** (87.1–93.9) | 81.99 (77.1–86.3) |

*Table 4.* Ablation study on SFT configurations (default teacher LLM: DeepSeek-V3.1).

| Factor | Configuration | Top-1 | Inf.Suff. |
|---|---|---|---|
| **Default** | Salus-Zero-7B | **55.02** | **65.80** |
| Data Scale | 50% Data | 54.28 | 64.31 |
| | 25% Data | 50.19 | 61.34 |
| | 12.5% Data | 44.61 | 62.45 |
| Model Size | 3B Size Variant | 43.87 | 60.60 |
| Teacher | DeepSeek-V3.2 | 43.12 | 62.82 |
| | GLM-Z1-Flash | 31.97 | 62.45 |

teacher model is pivotal; DeepSeek-V3.1 provides superior strategic supervision compared to DeepSeek-V3.2, which indicates that a greater teacher model (DeepSeek-V3.2 shows better performance in Table 2) not always leads to better performance for generating SFT data.

**Efficacy of Multi-agent RL.** Our results in Table 5 reveal a powerful *synergy* between the Differential Reasoner (DR) and Strategic Controller (SC). Optimizing both components (7B variant) elevates Top-1 accuracy from 55.02% (SFT) to 83.64%. In contrast, applying RL training solely to the DR only reaches 61.71%, underscoring the SC's essential role in strategically managing workup depth. This improvement aligns with clinical intuition: without a smart SC, LLMs frequently suffer from *premature closure*, as illustrated by

Table 4, the quality of the SFT phase is determined by data scale, model capacity, and teacher model quality. We observe a *positive correlation* between diagnostic performance and SFT data volume; increasing the scale from 12.5% to 100% yields a 10.41% absolute gain in Top-1 accuracy. Furthermore, model size is critical, as the 7B variant outperforms its 3B counterpart by 11.15%. Finally, the choice of

*Table 5.* Ablation study on `Salus` variants about RL.

| Variant | DR | SC | Top-1 | Inf.Suff. |
|---|---|---|---|---|
| `Salus-7B` | ✓ | ✓ | **83.64** | **72.86** |
| – w/o RL for SC | ✓ | – | 61.71 | 65.43 |
| – SFT only | – | – | 55.02 | 65.80 |
| `Salus-3B` | ✓ | ✓ | 65.06 | 65.06 |

the failure case in Figure 1. Through RL, the SC learns a stricter evidence-sufficiency boundary for deciding when to stop gathering evidence, thereby preventing premature termination and driving the performance gain in Table 5. Notably, the RL-optimized 3B model (65.06%) surpasses many open-source LLMs such as Qwen3-32B (43.12% in Table 2), demonstrating the effectiveness of our RL methods.

Due to page limit, additional experiments are deferred to Appendix A, including: 1) a qualitative **case study** demonstrating *Salus's superior strategic rigor in resisting premature closure* compared to DeepSeek-V3.2, GPT-5.2, and Gemini 3 Flash; 2) an evaluation of the DR's standalone diagnostic capacity for the **static clinical reasoning** task; 3) a comparative study of multi-agent versus single-agent architectures; and 4) a cross-model robustness evaluation where Gemini 3 Flash replaces DeepSeek-V3.1 as the Diagnostic Environment backbone. These results collectively verify the robustness and effectiveness of `Salus`.

## 5. Related Work

**LLMs for Sequential Medical Decision-Making.** Many recent works focus on the interactive and sequential decision-making. To simulate realistic clinical workflows, several frameworks have been proposed: Li et al. (2024) and Fan et al. (2025) established virtual hospital environments to evaluate agents on symptom collection and examination tasks, while Schmidgall et al. (2024) introduced *AgentClinic*, a multimodal benchmark requiring agents to operate under incomplete information. To navigate these complex settings, prior works have primarily relied on sophisticated prompt engineering and agent orchestration. For instance, Nori et al. (2025) designed a multi-agent orchestrator simulating a physician panel to optimize test selection, Liu et al. (2024) addressed the full pipeline from referral to treatment, and Qiu et al. (2025b) proposed metrics to quantify reasoning efficiency and completeness in multi-round contexts. Regarding model optimization, Tu et al. (2025) developed AMIE, which achieves expert-level conversational diagnosis via self-play but remains closed-source. Similarly, Qiu et al. (2025a) explored end-to-end multi-round RL with intermediate rewards. Despite these progress, existing approaches exhibit critical limitations: prompt-based agentic workflows (Nori et al., 2025; Liu et al., 2024) lack targeted

parameter updates and heavily rely on the inherent capacity of proprietary backbone models, while current open-source efforts often struggle to resolve high-complexity cases requiring deep strategic reasoning.

**Multi-Agent Frameworks for Clinical Reasoning.** Recent studies utilize multi-agent architectures to emulate clinical collaboration. A prominent paradigm involves simulating Multi-disciplinary Teams to enhance diagnostic accuracy. Tang et al. (2024) and Chen et al. (2024a) pioneered role-playing frameworks where agents engage in collaborative discussions to solve medical QA and rare disease cases, respectively. Similarly, Kim et al. (2024) introduced adaptive team structuring that dynamically assigns collaboration patterns based on case difficulty. Other approaches focus on functional specialization: Zhou et al. (2025) decomposes diagnosis into distinct roles (e.g., Radiologist, Director) to facilitate modular knowledge updates, while Liu et al. (2025) and Zhao et al. (2025) emphasize reliability through verifiable reasoning chains and multi-modal integration. Furthermore, Almansoori et al. (2025) incorporated self-improvement mechanisms, combining multi-agent debate with experience-based retrieval. However, these frameworks rely heavily on complex prompt engineering and static collaboration protocols, which imposes a strict dependency on the instruction-following capacities of the underlying backbone LLMs. Moreover, such intricate designs are not easy to transfer directly to the sequential diagnostic testing task.

## 6. Conclusion

In this paper, we presented `Salus`, a framework for navigating complex clinical diagnosis. By employing a multi-agent architecture and optimizing them through a reasoning-aware cold-start followed by multi-agent RL with step-wise rewards, we demonstrated that our 7B model achieves a SOTA Top-1 accuracy of 83.64% on high-complexity cases, outperforming substantially larger open-source models. Currently, `Salus` prioritizes diagnostic precision over resource constraints and its decision-making logic does not yet explicitly account for testing costs, which may limit its application in real clinical scenarios. Future research will focus on optimizing the trade-off between diagnostic accuracy and testing efficiency.

## Acknowledgements

This study was supported by grants from the National Science Foundation of China (T2541010), the National Key R&D Program of China (2024YFF1207100, 2024YFF1207103), and Beijing National Research Center for Information Science and Technology (BNRist). The funders had no roles in study design, data collection and analysis, publication decisions, or manuscript preparation.

## Impact Statement

This work investigates the application of multi-agent reinforcement learning to sequential medical diagnosis. While `Salus` demonstrates a superior capability to reason strategically in simulated environments, we emphasize that this research is a methodological exploration and is *not ready for clinical deployment*.

Several critical **limitations** regarding the `Salus` framework must be acknowledged. First, **modality constraints**: the current system operates solely on text-based clinical records. In actual practice, complex diagnosis heavily relies on interpreting raw multimodal data, such as radiological imaging, histopathology slides, and physiological signals, which are currently beyond the scope of this LLM-based architecture. Second, **cost-benefit modeling**: although the Strategic Controller (SC) is designed to mitigate over-testing, our reward structure does not yet explicitly quantify the financial costs, physical invasiveness (e.g., biopsy risks), or the time-sensitive nature of diagnostic procedures (e.g., the delay of lab cultures). Third, **environment fidelity**: while our *Diagnostic Environment* utilizes grounded clinical records, the use of an LLM as a proxy for the hospital infrastructure may lack the stochastic noise, false positives, and diagnostic ambiguities inherent in real-world laboratory testing.

Beyond technical limitations, broader ethical risks remain. Despite our reward engineering, LLMs are susceptible to hallucinations and may generate plausible yet factually incorrect medical advice. Furthermore, our training data, sourced from specific high-impact journals and online communities, may harbor inherent demographic or geographic biases, potentially leading to inequitable performance across diverse patient populations.

Therefore, `Salus` should be viewed strictly as a research prototype intended to advance the understanding of automated clinical reasoning. It is not a medical device. Any future development towards clinical decision support must undergo rigorous prospective clinical trials and strict regulatory approval. We advocate for a *human-in-the-loop* paradigm, where such systems serve solely as assistive tools for qualified medical professionals who retain full accountability for patient care.

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

# A. Additional Experiments

## A.1. Case Study: Navigating Complex Clinical Synthesis

To qualitatively evaluate the diagnostic rigor of Salus, we analyze a representative trajectory of a 73-year-old female presenting with systemic erythroderma and necrotic ulcers (Figure 4). This case exemplifies a complex clinical challenge: distinguishing primary malignancies from reactive inflammatory processes triggered by secondary infections.

**Frontier Models: The Pitfall of Premature Closure.** As illustrated in the full trajectory (Figure 4), at step $t - 1$, initial clinical findings reveal evidence of HSV-2 and *P. aeruginosa* infections. Prompted with this context, frontier models GPT-5.2, Gemini 3 Flash and DeepSeek-V3.2 (acting as the Strategic Controller, SC) both issue a Stop signal, accepting *Eczematoid Herpes* ($d^{(1)}$) as the final diagnosis. This exemplifies the classic pitfall of *premature closure*; while the diagnosis is *Partially Correct* as it identifies the immediate infection, it fails to account for the underlying etiological root—a malignancy-driven immune collapse.

**Salus Path: Strategic Prudence and Diagnostic Resolution.** In contrast, Salus recognizes that localized infections cannot fully explain systemic findings like extreme IgE elevation ($25, 113$ IU/mL) and hypereosinophilia. Guided by the RL-optimized SC, Salus elects to Continue for two additional rounds of investigation. This strategic persistence allows the model to navigate through layered evidence, eventually resolving the diagnostic complexity to reach Angioimmunoblastic T-cell Lymphoma (AITL) by step $t + 1$. The resulting diagnosis is categorized as *Completely Correct*, demonstrating Salus's ability to prioritize evidence sufficiency over testing cost in high-stakes scenarios.

**High-Resolution Clinical Synthesis by the DR at Step $t$.** Focusing on the granular behavior at step $t$ (Figure 5), the Differential Reasoner (DR) demonstrates superior synthesis capabilities. It correctly interprets the "immune collapse" pattern and ranks *AITL* as the primary hypothesis ($d^{(1)}$). Crucially, the DR maintains a high-resolution differential list that includes critical mimics like Stevens-Johnson Syndrome (SJS) ($d^{(4)}$) and Primary Immune Dysregulation ($d^{(5)}$). This ensures that subsequent decision-making is grounded in a comprehensive belief set, preventing the model from ignoring low-probability but high-risk etiologies.

**Hypothesis-Driven Testing by the WP at Step $t$.** The Workup Proposer (WP) further validates the model's clinical expertise by proposing hypothesis-driven investigations rather than routine broad-spectrum panels. As detailed in Figure 5, the WP specifically requests:

- **FDG PET-CT & Bone Marrow Biopsy:** Requested for systemic staging and to evaluate potential bone marrow involvement, which is critical for confirming the extent of the malignant AITL hypothesis ($d^{(1)}$).

- **TCR/IgH Dual Rearrangement Analysis:** Utilized to identify AITL-specific B-cell clonal expansion, providing the molecular evidence required to confirm the primary diagnosis of $d^{(1)}$.

- **Quantitative Peripheral TCR qPCR:** A high-precision investigation designed to distinguish malignant T-cell proliferation from reactive expansions triggered by secondary infections, such as Eczema Herpeticum ($d^{(2)}$) or polymicrobial superinfection ($d^{(3)}$).

- **Serum Immunofixation & Lymphocyte Proliferation Assay:** Employed to assess B-cell dyscrasias and T-cell functional integrity, serving to systematically rule out primary immune dysregulation syndromes ($d^{(5)}$).

- **Serum Anti-T-cell Antibody Assay:** Specifically targeted at investigating drug-induced immune triggers to provide the "negative evidence" necessary to definitively exclude critical mimics like Stevens-Johnson Syndrome ($d^{(4)}$).

These targeted requests demonstrate that the WP understands the *discriminatory value* of each test in resolving specific differential uncertainties.

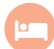

**Medical History:**
A 73-year-old Black female with chronic eczema and psoriasis presented with perianal pain 8 days ago. After initial antibiotic treatment, she developed a rapidly spreading rash in the groin 2 days post-discharge. It progressed to confluent, "punched-out" ulcers with purulence on the lower abdomen, perineum, and inner thighs. Over 90% of her skin showed faint erythema. Outside hospital evaluation noted significant eosinophilia. She has a history of type 2 diabetes, hypertension, and genital herpes (last episode 3 years ago).

**Physical Exam:**
Vital signs are stable. BMI is low at 17.2 kg/m². Exam shows confluent, monomorphic "punched-out" ulcerated plaques with foul-smelling discharge in axillae, groin, lower abdomen, and skin folds. Over 90% BSA has faint blanching erythema. She has nail dystrophy, scaly fissures on hands, and non-tender lymphadenopathy. No mucosal involvement.

**Test Results:**
- Microbiology: Skin swab positive for HSV-2 and Pseudomonas aeruginosa (multi-sensitive, including piperacillin-tazobactam and carbapenems). Blood cultures confirmed P. aeruginosa bacteremia within 24h. Peripheral blood PCR showed elevated EBV DNA (3,500 IU/mL); HHV-6 and CMV were negative.
- Laboratory Findings: Significant leukocytosis (WBC 19,700/μL; elevated neutrophils, lymphocytes, and eosinophils), anemia (Hgb 11.2 g/dL), and markedly elevated inflammatory markers (CRP 130.9 mg/L, ESR 98 mm/h, PCT 35.2 ng/mL). Comprehensive metabolic panel and coagulation studies showed mild hepatic/renal impairment and prolonged PT/APTT. Serum IgE was severely elevated (25,113 IU/mL).
- Pathology: Skin biopsy revealed focal epidermal necrosis, ballooning degeneration, and multinucleated giant cells (consistent with herpes infection), plus a dermal infiltrate rich in eosinophils. Direct immunofluorescence (DIF) was normal.
- Imaging: Pelvic MRI showed extensive perianal/perineal soft tissue edema with enhancement, a 1.8x1.5 cm post-anal abscess, and bilateral inguinal/iliac lymphadenopathy.

**$D_{t-1}$:**
$d^{(1)}$: Eczematoid Herpes (Kaposi Varicelliform Eruption)/Generalized Herpes Simplex Virus Type 2 Skin Infection, combined with secondary Pseudomonas aeruginosa skin and soft tissue infection and bacteremia, coexisting with perianal abscess
$d^{(2)}$: Perineal Necrotizing Fasciitis (Fournier Gangrene), possibly disseminated from perianal infection and complicated by Pseudomonas aeruginosa invasive infection/bacteremia
$d^{(3)}$: Pseudomonas aeruginosa Pyoderma Gangrenosum (Ecthyma Gangrenosum), combined with Pseudomonas aeruginosa bacteremia
$d^{(4)}$: Drug-Induced Stevens-Johnson Syndrome/Toxic Epidermal Necrolysis
$d^{(5)}$: Drug Reaction with Eosinophilia and Systemic Symptoms (DRESS syndrome)

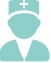

SC: GPT-5.2 & Gemini 3 Flash & DeepSeek-V3.2
**$s_{t-1}$: Stop**
Submit $d^{(1)}$= Eczematoid Herpes (Kaposi Varicelliform Eruption)… as final diagnosis: **Partially Correct**

---

**Trajectories of Salus**

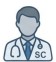

While there are some points supporting each diagnosis, **none can fully explain all the findings with definitive evidence**.
**$s_{t-1}$: Continue**

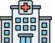

**Subsequent tests $E_{t-1}$ and the corresponding results:**
- Skin & Lymph Node Biopsies: Dermal and nodal infiltration by atypical lymphocytes; IHC: CD3+/CD4+/CD10+/BCL6+; TCR rearrangement: Clonal.
- Microbiology (mNGS/Culture/PCR): Confirmed polymicrobial infection with HSV-1, Pseudomonas aeruginosa, and Bacteroides fragilis.
- CBC & Serology: Marked eosinophilia (AEC 6,820/μL), elevated ESR (80 mm/h), and extreme IgE elevation (25,113 IU/mL).
- Flow Cytometry: Identified an abnormal T-cell population (20% of lymphocytes).
- Abdominal & Pelvic Ultrasound: Perianal abscess (1.8x1.5 cm) and extensive inguinal/iliac lymphadenopathy.

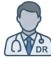

**$D_t$:**
- $d^{(1)}$: Angioimmunoblastic T-cell Lymphoma (AITL)
- $d^{(2)}$: Eczema Herpeticum (secondary to HSV-1)
- $d^{(3)}$: Polymicrobial Superinfection
- $d^{(4)}$: Stevens-Johnson Syndrome (SJS)
- $d^{(5)}$: Primary Immune Dysregulation (e.g., Hyper-IgE Syndrome)

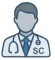

**Out of clinical prudence**, continue the session to exclude critical mimics like SJS.
**$s_t$: Continue**

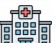

**Subsequent tests $E_t$ and the corresponding results:**
- FDG PET-CT & Bone Marrow Biopsy: PET-CT showed widespread hypermetabolic lymphadenopathy (SUVmax 12.5), skin ulcers (SUVmax 8.7), and pulmonary nodules (SUVmax 10.2). Bone marrow biopsy revealed 30% abnormal lymphocyte infiltration.
- TCR/IgH Dual Rearrangement Analysis: Skin and lymph node specimens showed clonal TCR rearrangement and normal polyclonal IgH pattern.
- Quantitative Peripheral TCR qPCR: Detected clonal T-cell populations in peripheral blood.
- Serum Immunofixation & Lymphocyte Proliferation Assay: Immunofixation and free light chain levels were normal. Lymphocyte proliferation assay showed significantly impaired T-cell response to PHA (stimulation index 9 ↓).
- Serum Anti-T-cell Antibody Assay: Negative.

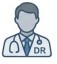

**$D_{t+1}$:**
- $d^{(1)}$: Angioimmunoblastic T-cell Lymphoma (AITL)
- $d^{(2)}$: Infection-triggered Reactive Lymphoproliferative Disorder
- $d^{(3)}$: Stevens-Johnson Syndrome (SJS) / Toxic Epidermal Necrolysis (TEN)
- $d^{(4)}$: Hypereosinophilic Syndrome (HES) with Hyper-IgE
- $d^{(5)}$: Pyoderma Gangrenosum (PG) Syndrome

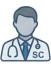

**$s_{t+1}$: Stop**
Submit $d^{(1)}$= Angioimmunoblastic T-cell Lymphoma (AITL) as final diagnosis: **Completely Correct**

*Figure 4.* The full diagnostic trajectory comparison. While GPT-5.2 and Gemini 3 Flash succumb to premature closure at step $t-1$ with a Partially Correct diagnosis, Salus successfully navigates the complex evidence to reach a Completely Correct diagnosis by step $t+1$.

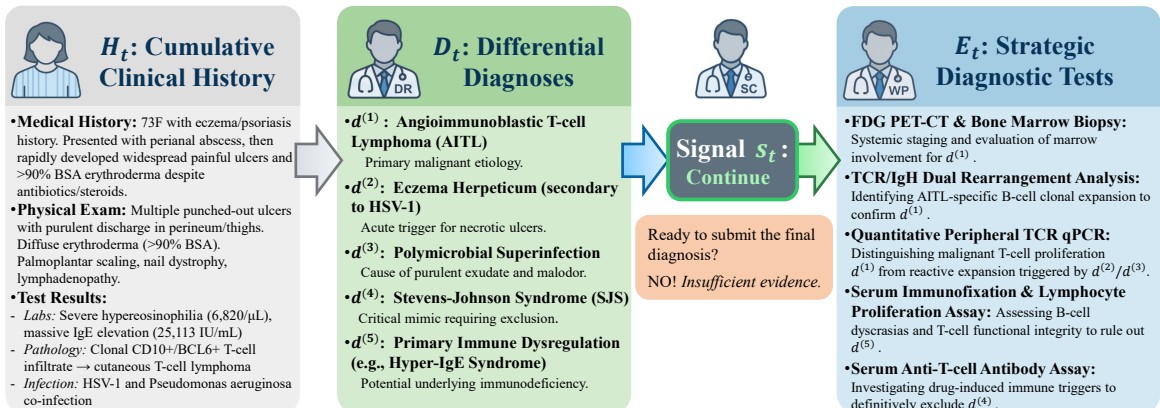

*Figure 5.* A detailed snapshot of Salus's decision-making at step $t$. The model identifies AITL as the primary etiology but elects to continue the session to exclude mimics. The WP proposes a targeted workup, where each test is explicitly linked to resolving specific diagnostic hypotheses ($d^{(1)}$–$d^{(5)}$).

## A.2. Additional Ablation Studies

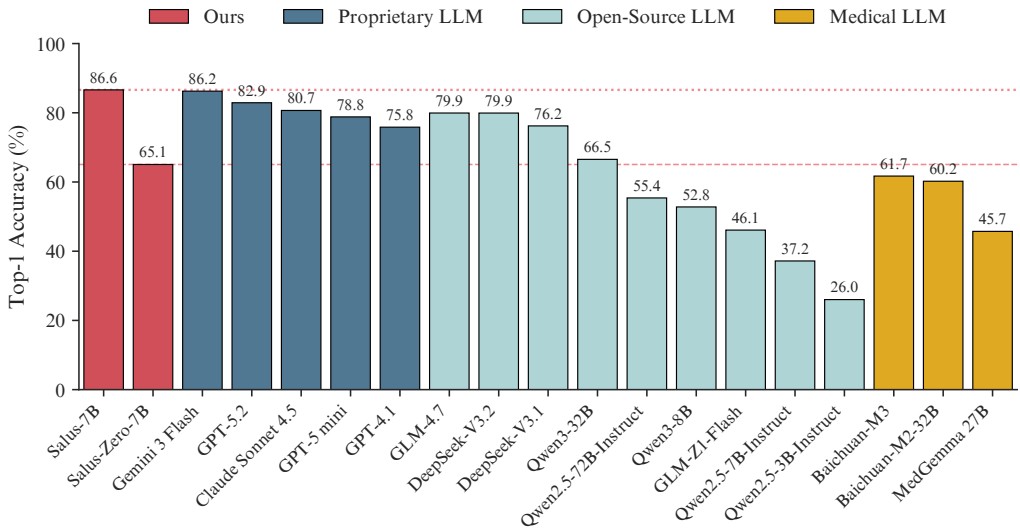

*Figure 6.* Comparison of **static** clinical reasoning performance on 269 complex diagnoses. Models are evaluated on their ability to derive the correct diagnosis given the complete clinical transcript (initial context plus all diagnostic test results documented in the record) in a single-round inference setting.

**Evaluation of Static Clinical Reasoning.** To isolate the diagnostic reasoning capability of the *Differential Reasoner* (DR) from its interaction strategy, we evaluate its performance in a static setting. In this configuration, the DR is provided with the complete patient record—encompassing both the initial clinical context and the full suite of diagnostic test results documented in the original patient record—while the gold-standard diagnosis is withheld. The Top-1 accuracy results are illustrated in Figure 6 and Table 6. Salus-7B achieves the highest accuracy, exhibiting diagnostic reasoning capabilities comparable to frontier models such as Gemini 3 Flash. Furthermore, Salus-Zero-7B outperforms all specialized medical LLMs and remains competitive with substantially larger general-purpose models like Qwen3-32B. These findings further verify the effectiveness of our reasoning-aware SFT and multi-agent RL optimization.

**Comparison with Existing Multi-Agent Systems.** We further compare Salus with representative medical multi-agent frameworks, including MedAgents, MDAgents, and DeepRare. Since these systems are primarily formulated for single-round

*Table 6.* **Comprehensive** comparison of **static** clinical reasoning performance on 269 complex diagnoses. The diagnostic quality is adjudicated based on the four-level rubric: Level 3 (Completely Correct), Level 2 (Clinically Acceptable), Level 1 (Partially Correct), and Level 0 (Incorrect). We report Top-1/2/3 diagnostic accuracies and the cumulative distribution of quality levels for the Top-1 prediction.

| LLM | Top-$k$ Accuracy ↑ | | | Four-level Distribution (%) | | | |
| --- | --- | --- | --- | --- | --- | --- | --- |
| | Top-1 | Top-2 | Top-3 | 3 | $\geq 2$ | $\geq 1$ | 0 ↓ |
| Salus-7B | **86.61** | 90.71 | **94.05** | 35.31 | **86.61** | 95.91 | 4.09 |
| Gemini 3 Flash | 86.25 | **91.45** | 91.82 | 45.35 | 86.25 | **97.39** | **2.60** |
| gpt-5.2 | 82.90 | 89.22 | 90.34 | **45.73** | 82.90 | 95.54 | 4.46 |
| claude-sonnet-4.5 | 80.67 | 88.11 | 90.34 | 32.34 | 80.67 | 95.17 | 4.84 |
| GLM-4.7 | 79.93 | 86.25 | 87.73 | 34.95 | 79.93 | 94.43 | 5.58 |
| DeepSeek-V3.2 | 79.92 | 86.25 | 87.73 | 37.55 | 79.92 | 95.91 | 4.09 |
| gpt-5-mini | 78.81 | 86.25 | 86.99 | 40.89 | 78.81 | 93.68 | 6.32 |
| DeepSeek-V3.1 | 76.21 | 82.90 | 85.13 | 32.34 | 76.21 | 92.94 | 7.06 |
| gpt-4.1 | 75.84 | 81.78 | 84.01 | 33.08 | 75.84 | 92.56 | 7.44 |
| Qwen3-32B | 66.54 | 71.75 | 74.72 | 18.96 | 66.54 | 90.70 | 9.29 |
| Salus-Zero-7B | 65.06 | 71.00 | 74.72 | 23.05 | 65.06 | 89.60 | 10.41 |
| Baichuan-M3 | 61.71 | 67.66 | 70.63 | 27.14 | 61.71 | 88.85 | 10.78 |
| Baichuan-M2-32B | 60.22 | 67.66 | 72.12 | 22.68 | 60.22 | 83.64 | 16.36 |
| Qwen2.5-72B-Instruct | 55.39 | 62.83 | 66.54 | 17.84 | 55.39 | 85.50 | 14.50 |
| Qwen3-8B | 52.79 | 58.36 | 63.57 | 12.64 | 52.79 | 80.67 | 19.33 |
| GLM-Z1-Flash | 46.10 | 52.05 | 56.50 | 11.52 | 46.10 | 77.69 | 22.31 |
| medgemma-27b-it | 45.73 | 52.42 | 56.13 | 9.29 | 45.73 | 70.26 | 29.74 |
| Qwen2.5-7B-Instruct | 37.18 | 44.98 | 48.33 | 8.92 | 37.18 | 68.03 | 31.97 |
| Qwen2.5-3B-Instruct | 26.02 | 32.34 | 37.92 | 6.32 | 26.02 | 59.85 | 39.78 |

*Table 7.* Comparison with representative medical multi-agent systems on 269 high-complexity cases in the static diagnosis setting. All baseline systems are implemented with `DeepSeek-V3.2` as the backbone LLM.

| Framework | Paradigm | Top-1 | Top-2 | Top-3 |
| --- | --- | --- | --- | --- |
| MedAgents (Tang et al., 2024) | Multi-expert collaboration | 71.7 | 79.6 | 84.0 |
| MDAgents (Kim et al., 2024) | Adaptive collaboration | 71.7 | 83.6 | 86.6 |
| DeepRare (Zhao et al., 2025) | Autonomous rare disease team | 78.4 | 82.5 | 85.5 |
| Salus-7B | DR-SC-WP | **86.6** | **90.7** | **94.1** |

diagnostic reasoning, we evaluate all methods in the static setting described above: each model receives the complete clinical record and predicts the final diagnosis without additional interaction. As shown in Table 7, `Salus-7B` consistently achieves higher Top-1/2/3 accuracy. This result indicates that our proposed method improves beyond generic role specialization, as the RL-optimized reasoning and control modules provide stronger support for complex diagnostic synthesis.

**Effect of Multi-Agent Decomposition.** We further investigate the impact of the proposed multi-agent framework by comparing it against a standard single-agent baseline, where a unified model prompt handles reasoning, termination, and workup proposals simultaneously. As summarized in Table 8, frontier models such as `Gemini 3 Flash` and `DeepSeek-V3.2` derive substantial benefits from the multi-agent architecture, exhibiting absolute Top-1 accuracy gains of 6.33% and 4.47%, respectively.

This performance uplift suggests that decoupling the high-stakes cognitive tasks of *differential synthesis* (DR) and *termination control* (SC) alleviates the instruction-following burden and prevents the model from conflating diagnostic reasoning with administrative decision-making. Interestingly, while the single-agent variants of `Gemini` and `GPT-5-mini` occasionally achieve higher *Information Sufficiency* (Inf. Suff.), they fail to translate this additional information into higher diagnostic accuracy. This indicates that for frontier LLMs, the diagnostic bottleneck is not merely information elicitation, but the structured synthesis of gathered evidence. Conversely, we observe that the mid-sized `Qwen3-32B` model performs better in the single-agent setting. This may be attributed to the inherent difficulty of small-to-mid-sized models in managing the specialized, complex role-play instructions required by the tripartite architecture, suggesting a potential trade-off between architectural complexity and the base model's instruction-following capacity.

We also provide a comparison between the proposed multi-agent training pipeline and a single-agent baseline trained with the same data and backbone model. This single-agent baseline uses one unified policy to produce diagnoses, make termination

*Table 8.* Performance Comparison between Multi-agent and Single-agent Frameworks on 269 High-Complexity Cases. The multi-agent approach generally enhances diagnostic precision in frontier models by decoupling reasoning from strategic control.

| Framework | Backbone LLM | Accuracy ↑ | | | Inf. Suff. ↑ | Efficiency Metrics | |
|---|---|---|---|---|---|---|---|
| | | Top-1 | Top-2 | Top-3 | | #Rounds | #Tests |
| Multi-agent | Gemini 3 Flash | **79.93** | **84.39** | **86.62** | 72.12 | 0.85 | 8.26 |
| Single-agent | Gemini 3 Flash | 73.60 | 76.95 | 78.07 | **73.61** | 1.34 | 7.15 |
| Multi-agent | DeepSeek-V3.2 | **71.38** | **77.32** | **78.44** | **66.91** | 1.88 | 11.95 |
| Single-agent | DeepSeek-V3.2 | 66.91 | 70.26 | 72.49 | 66.17 | 1.82 | 7.85 |
| Multi-agent | GPT-5-mini | **68.40** | **75.09** | **76.95** | 63.57 | 1.06 | 14.82 |
| Single-agent | GPT-5-mini | 67.29 | 69.89 | 73.98 | **67.66** | 1.13 | 8.98 |
| Multi-agent | Qwen3-32B | 43.12 | 49.81 | 53.53 | 52.05 | 1.11 | 7.00 |
| Single-agent | Qwen3-32B | **53.16** | **58.00** | **60.97** | **63.20** | 2.40 | 9.77 |

*Table 9.* Comparison between the proposed multi-agent training pipeline and a single-agent baseline trained with the same data and backbone model. `Qwen2.5-7B-Instruct` is selected as the backbone LLM.

| Model | Top-1 | Inf. Suff. | #Rounds |
|---|---|---|---|
| `Salus-7B` (Multi-agent SFT + RL) | **83.6** | **72.9** | 2.9 |
| `Salus-Zero-7B` (Multi-agent SFT) | 55.0 | 65.8 | 2.5 |
| Single-agent (SFT + RL) | 53.5 | 67.1 | 3.0 |
| Single-agent (SFT) | 53.5 | 63.6 | 3.2 |

decisions, and request additional tests as described above. And it is trained under a standard multi-round RL Environment, where the reward is only given when reaching the final diagnosis. As shown in Table 9, RL yields only marginal gains for the single-agent formulation. A plausible explanation is reward sparsity: the unified policy receives task-level feedback only at the end of a long trajectory, whereas the decomposed `Salus` architecture exposes role-specific intermediate states and enables denser supervision for the DR and SC. These results indicate that the improvement of `Salus` relies on the interaction between architectural decomposition and role-aligned optimization signals.

**Robustness across Diagnostic Environments.** To ensure that our evaluation is not biased toward a specific model family, we conduct a cross-model robustness check by replacing the `DeepSeek-V3.1` backbone with `Gemini 3 Flash` to serve as the *Diagnostic Environment*, the *Judge*, and the *Oracle*. As summarized in Table 10, the relative performance rankings remain highly consistent with our primary results in Table 2. Specifically, `Salus-7B` continues to achieve state-of-the-art performance, attaining a Top-1 accuracy of 86.62%, comparable to `GPT-5.2` (86.25%) and surpassing `Gemini 3 Flash` (85.13%) and `DeepSeek-V3.2` (81.78%). While absolute scores exhibit slight variations due to differences in adjudication strictness across models, the sustained superiority of `Salus-7B` across different evaluation infrastructures validates that its strategic reasoning and diagnostic capabilities are robust and agnostic to the choice of the underlying environment model.

*Table 10.* **Cross-model robustness evaluation on 269 high-complexity cases.** This table reports diagnostic performance where `Gemini 3 Flash` (instead of `DeepSeek-V3.1`) serves as the *Diagnostic Environment*, *Judge*, and *Oracle*. **Top-1/2/3 Accuracy** and **Inf. Suff.** quantify diagnostic precision and evidence completeness, while #Turns and #Tests measure interaction efficiency. Results demonstrate consistent SOTA performance of `Salus-7B` across different evaluation infrastructures. Best results are **bolded**; 95% CIs are reported for key metrics.

| LLM | Top-1 (95% CI) | Top-2 | Top-3 | Inf. Suff. ↑ (95% CI) | #Rounds | #Tests |
|---|---|---|---|---|---|---|
| `Salus-7B` | **86.62** (76.6–93.3) | 88.85 | 91.08 | **89.96** (80.5–95.5) | 2.63 | 18.77 |
| GPT-5.2 | 86.25 (74.7–92.2) | **91.45** | **91.82** | 89.59 (80.5–95.5) | 1.55 | 25.39 |
| Gemini 3 Flash | 85.13 (74.7–92.2) | 87.73 | 88.85 | 85.13 (74.7–92.2) | 0.90 | 8.84 |
| DeepSeek-V3.2 | 81.78 (71.0–89.8) | 86.62 | 87.36 | 85.13 (74.7–92.2) | 1.79 | 11.43 |
| DeepSeek-V3.1 | 74.35 (62.1–83.4) | 80.67 | 82.16 | 87.36 (76.6–93.3) | 1.89 | 9.98 |

**Comprehensive Evaluation on Routine Clinical Cases.** Table 11 provides a multi-dimensional comparison of diagnostic performance on 483 routine cases, encompassing accuracy, information elicitation, and interaction efficiency. Consistent with the trends observed in high-complexity scenarios, `Salus-7B` maintains its state-of-the-art (SOTA) status across all diagnostic precision metrics, achieving a Top-1 accuracy of 91.10% and a Top-3 accuracy of 95.45%. This performance

*Table 11.* **Comprehensive** diagnostic performance on 483 routine clinical cases. Salus-7B achieves consistent SOTA performance, indicating strong generalization across different levels of clinical complexity. Best results are **bolded**.

| LLM | Top-1 (95% CI) | Top-2 | Top-3 | Inf. Suff. ↑ (95% CI) | #Rounds | #Tests |
|---|---|---|---|---|---|---|
| Salus-7B | **91.10** (87.1–93.9) | **94.62** | **95.45** | 81.99 (77.1–86.3) | 2.52 | 14.37 |
| Gemini 3 Flash | 88.20 (83.7–91.4) | 91.93 | 92.34 | **84.06** (79.2–87.9) | 0.98 | 8.83 |
| GPT-5.2 | 86.34 (81.6–89.9) | 89.65 | 90.68 | 80.75 (75.5–84.9) | 1.69 | 19.07 |
| DeepSeek-V3.2 | 82.19 (77.1–86.3) | 87.58 | 88.82 | 82.82 (77.9–86.9) | 2.04 | 10.64 |
| DeepSeek-V3.1 | 81.57 (76.3–85.6) | 85.71 | 86.96 | 81.57 (76.3–85.6) | 2.56 | 8.60 |
| GPT-5 mini | 80.95 (75.9–85.3) | 86.75 | 89.23 | 83.85 (79.2–87.9) | 0.90 | 9.61 |
| Salus-Zero-7B | 78.67 (73.5–83.3) | 83.85 | 85.51 | 83.23 (78.3–87.3) | 2.17 | 8.56 |
| Qwen3-32B | 76.19 (70.8–80.9) | 82.40 | 83.85 | 76.19 (70.8–80.9) | 0.83 | 4.45 |
| MedGemma 27B | 75.16 (69.6–79.9) | 80.33 | 82.40 | 80.95 (75.9–85.3) | 1.32 | 9.35 |
| Qwen2.5-32B-Instruct | 72.67 (66.9–77.5) | 78.88 | 82.19 | 81.16 (75.9–85.3) | 1.09 | 5.28 |
| Qwen3-8B | 66.87 (61.1–72.3) | 73.91 | 75.98 | 76.40 (71.2–81.2) | 1.31 | 6.77 |
| Qwen2.5-7B-Instruct | 59.63 (53.5–65.2) | 65.84 | 68.74 | 80.95 (75.9–85.3) | 1.95 | 11.22 |
| Baichuan-M2-32B | 58.18 (52.4–64.1) | 61.08 | 62.11 | 62.73 (56.9–68.4) | 1.72 | 12.86 |

not only surpasses frontier proprietary models like Gemini 3 Flash and GPT-5.2 but also demonstrates a substantial generalization capability, as the model effectively scales its reasoning logic from rare, complex reports to common primary care encounters.

In terms of efficiency, while models such as Qwen3-32B exhibit minimal testing rounds (#Rounds = 0.83), their lower diagnostic accuracy suggests a tendency toward premature closure. In contrast, Salus-7B adopts a more thorough diagnostic strategy (#Rounds = 2.52), mirroring a prudent clinical approach that prioritizes evidentiary support over the cost of diagnostic tests. Furthermore, the sustained improvement of Salus-7B over its SFT baseline (+12.43% Top-1 accuracy) reinforces the robustness of our reinforcement learning framework in calibrating strategic behavior across the full spectrum of clinical complexity.

**Evaluation on OOD test set.** We report the performance on 69 high-complex cases from JTO journal, which is an OOD test set. The results in Table 12 show similar ranking compared to the results on 269 cases in Table 2.

*Table 12.* Diagnostic performance on 62 **OOD** cases from journal JTO. Similar ranking is observed and Salus-7B achieves consistent SOTA performance. Best results are **bolded**.

| LLM | Top-1 (95% CI) | Top-2 | Top-3 | Inf. Suff. ↑ (95% CI) | #Rounds | #Tests |
|---|---|---|---|---|---|---|
| Salus-7B | **85.48** (74.7–92.2) | **87.10** | **87.10** | **75.81** (63.8–84.8) | 1.95 | 11.00 |
| Gemini 3 Flash | 80.65 (69.1–88.6) | 85.48 | **87.10** | 69.35 (57.0–79.4) | 0.68 | 6.16 |
| GPT-5.2 | 77.42 (65.6–86.0) | 79.03 | 79.03 | 74.19 (62.1–83.4) | 1.89 | 25.50 |
| Claude Sonnet 4.5 | 70.97 (58.7–80.8) | 74.19 | 74.19 | 72.58 (60.4–82.1) | 1.11 | 12.27 |
| GPT-5 mini | 70.97 (58.7–80.8) | 75.81 | 77.42 | 62.90 (50.5–73.8) | 0.85 | 11.02 |
| DeepSeek-V3.1 | 66.13 (53.7–76.7) | 67.74 | 74.19 | 66.13 (53.7–76.7) | 2.11 | 7.90 |
| DeepSeek-V3.2 | 64.52 (52.1–75.3) | 72.58 | 72.58 | 69.35 (57.0–79.4) | 1.60 | 7.97 |
| GLM-4.7 | 61.29 (48.8–72.4) | 67.74 | 69.35 | 64.52 (52.1–75.3) | 0.98 | 6.45 |
| Salus-Zero-7B | 58.06 (45.7–69.5) | 64.52 | 69.35 | 67.74 (55.4–78.0) | 2.10 | 9.65 |
| Qwen3-32B | 56.45 (44.1–68.1) | 66.13 | 69.35 | 53.23 (41.0–65.1) | 0.82 | 4.44 |
| GPT-4.1 | 54.84 (42.5–66.6) | 66.13 | 69.35 | 51.61 (39.4–63.6) | 0.77 | 4.19 |
| Qwen2.5-72B-Instruct | 51.61 (39.4–63.6) | 61.29 | 62.90 | 53.23 (41.0–65.1) | 0.89 | 5.44 |
| MedGemma 27B | 50.00 (37.9–62.1) | 59.68 | 61.29 | 58.06 (45.7–69.5) | 1.18 | 8.74 |
| Baichuan-M2-32B | 50.00 (37.9–62.1) | 53.23 | 56.45 | 61.29 (48.8–72.4) | 2.42 | 18.47 |
| Baichuan-M3 | 46.77 (34.9–59.0) | 51.61 | 53.23 | 66.13 (53.7–76.7) | 1.23 | 4.92 |
| Qwen2.5-7B-Instruct | 43.55 (31.9–55.9) | 48.39 | 50.00 | 67.74 (55.4–78.0) | 2.32 | 15.69 |
| Qwen3-8B | 41.94 (30.5–54.3) | 51.61 | 53.23 | 58.06 (45.7–69.5) | 1.42 | 7.68 |
| GLM-Z1-Flash | 37.10 (26.2–49.5) | 48.39 | 56.45 | 64.52 (52.1–75.3) | 1.98 | 10.88 |
| Qwen2.5-3B-Instruct | 29.03 (19.2–41.3) | 35.48 | 38.71 | 64.52 (52.1–75.3) | 2.16 | 11.27 |

# B. Statistics and Characterization of CompDiag-Bench

To ensure a comprehensive evaluation of AI clinicians across diverse clinical scenarios, we analyze **CompDiag-Bench** from three dimensions: data source, specialties and diagnostic difficulty.

## B.1. Data Source Distribution

Table 13 provides a comprehensive breakdown of the multi-source patient records used in our study. To ensure a rigorous evaluation of advanced clinical reasoning, the high-complexity test set is exclusively curated from prestigious, high-impact clinical journals, including the *New England Journal of Medicine* (NEJM), the *Journal of the American Medical Association* (JAMA), and the *Journal of Thoracic Oncology* (JTO). Notably, all cases from JTO are held out entirely from the training corpus to serve as a strictly independent *Out-of-Distribution (OOD)* evaluation, testing the model's generalization capabilities across distinct clinical specialties and reporting styles. In addition to case reports, we also extract records from a open Medical Community *iiYi* [1].

*Table 13.* Data distribution across multi-source patient records.The columns **#train** and **#test** represent the total count of unique patient records allocated to the train and test sets, respectively. Notably, JTO records are reserved exclusively for the OOD test set.

| Source | #Train | #Test |
|---|---|---|
| the New England Journal of Medicine (NEJM) | 767 | 123 |
| the Journal of the American Medical Association (JAMA) | 1557 | 84 |
| the Journal of Thoracic Oncology (JTO) | 0 | 62 |
| the Annals of Internal Medicine: Clinical Cases (AIMCC) | 229 | 0 |
| the AME Publishing Company (AME) | 605 | 0 |
| the ACG Case Reports Journal (ACG) | 1268 | 0 |
| the Scholars Journal of Medical Case Reports (SJMCR) | 2544 | 0 |
| Heliyon | 306 | 0 |
| iiYi Medical Community | 1369 | 483 |

## B.2. Diagnostic Difficulty Taxonomy

We formalize a five-level rubric to categorize the diagnostic complexity of each patient record, reflecting the intensity of clinical synthesis and the necessity of advanced investigations:

- **Level 1 (Routine):** Cases with explicit chief complaints and pathognomonic presentations (e.g., acute pharyngitis). Diagnosis is typically reached through routine physical examinations.

- **Level 2 (Straightforward):** Cases requiring clarification of history (e.g., abdominal pain localization) and primary laboratory or imaging tests (e.g., blood counts, ultrasonography).

- **Level 3 (Moderate Complexity):** Cases presenting with non-specific symptoms (e.g., chronic fatigue) or overlapping clinical features. Diagnosis requires targeted investigations (e.g., metabolic panels, endocrine assays) and systemic differential reasoning.

- **Level 4 (High Complexity):** Cases involving rare clinical entities or atypical manifestations (e.g., paraneoplastic syndromes). These require advanced modalities (e.g., MRI, histopathology) and multidisciplinary synthesis to resolve diagnostic ambiguity.

- **Level 5 (Exceptional Complexity):** High-stakes cases with multi-system involvement or obscure mechanisms (e.g., rare genetic disorders). Diagnosis necessitates high-cost or high-risk investigations (e.g., whole-exome sequencing) and aligns with the frontiers of clinical research.

DeepSeek-V3.1 is prompted to classify each record based on the above rubric, and the difficulty distribution of the test set is reported in Figure 7, highlighting the benchmark's focus on challenging scenarios (Level 4–5), which predominantly originate from specialized case reports.

---

[1] https://bingli.iiyi.com/

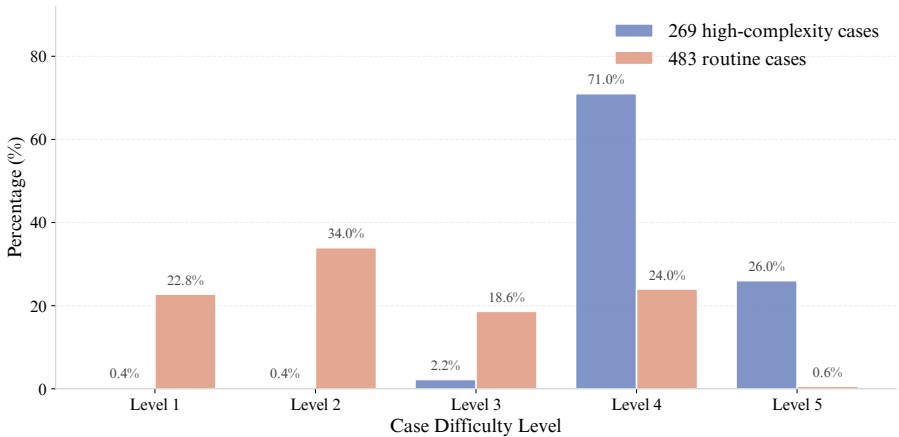

*Figure 7.* Distribution of Diagnostic Difficulty across the test set.

## B.3. Specialty Distribution

As illustrated in Figure 8, the benchmark encompasses a broad spectrum of medical sub-specialties, including Oncology, Neurology, and Cardiology, among others. This diversity ensures that the evaluated models are tested for generalized clinical reasoning rather than being overfitted to specific disease categories.

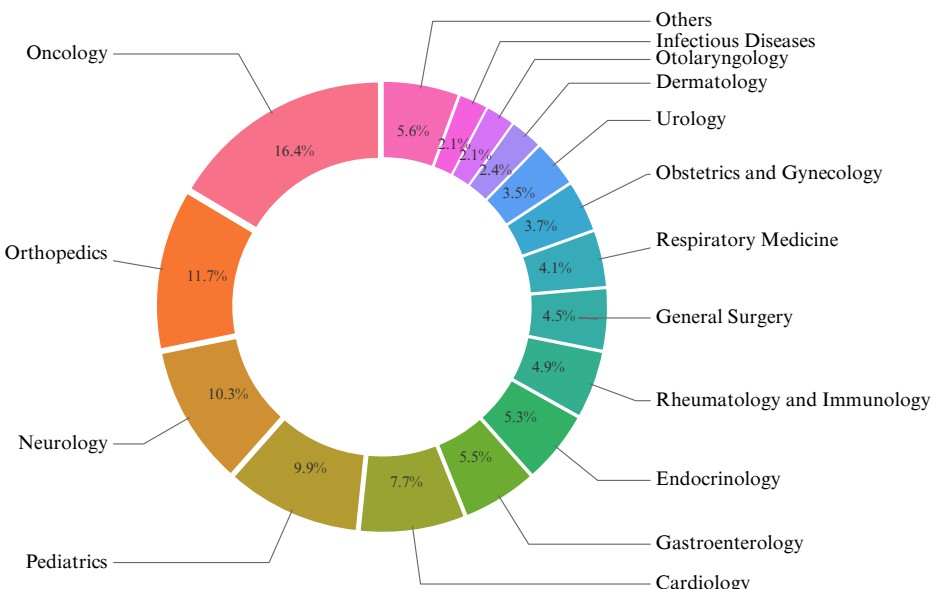

*Figure 8.* Specialty Distribution of cases in **CompDiag-Bench**.

## B.4. Data Contamination Risk Assessment

We assess potential data contamination by stratifying performance on the 269 high-complexity cases according to publication year. As shown in Table 14, `Salus-7B` maintains stable performance on cases published after the backbone model's knowledge cutoff (December 2023). Combined with its consistent performance on the held-out JTO OOD subset in Table 12,

*Table 14.* Multi-round diagnostic testing performance of `Salus-7B` on 269 high-complexity cases stratified by **publication year**. Top-1 Accuracy is reported.

| Publication Year | #Cases | Top-1 |
|---|---|---|
| 2025 | 61 | 86.9 |
| 2024 | 97 | 81.4 |
| ≤2023 | 111 | 83.8 |

this analysis suggests that the model's gains are unlikely to be explained by memorization of public case reports alone.

### B.5. Human Verification and Filtering Effects

To ensure the clinical validity of **CompDiag-Bench**, the test set is reviewed by 15 physicians with an average of more than 10 years of clinical experience. We use a cascade filtering protocol: candidate cases are first screened by two LLM filters (`DeepSeek-V3.1` and `GPT-5-mini`) and are then manually verified by clinical experts. The expert agreement rate reaches $93.4\%$, supporting the reliability of the final retained set. By contrast, the two LLM filters exhibit only fair agreement (Cohen's $\kappa = 0.25$), indicating that automated screening is useful for reducing the candidate pool but should not replace expert adjudication.

*Table 15.* Filtering effects across data sources. "After LLM" denotes cases retained after automated filtering but before human verification.

| Source | Before LLM Filter | After LLM | Final Retained |
|---|---|---|---|
| NEJM | 131 | 128 | 123 |
| JTO | 290 | 70 | 62 |
| JAMA | 124 | 86 | 84 |
| iiYi | 521 | 521 | 483 |

### B.6. An Example Record

Figure 9 illustrates a representative organized patient record within **CompDiag-Bench**. Each record is standardized via the information extraction pipeline detailed in Figure 3(a), which reformats disorganized clinical narratives into a consistent architecture of clinical history, physical examination findings, and diagnostic test results. Notably, to mirror the complexity of real-world medicine, the initial clinical context provided to the AI clinician may encompass longitudinal data, such as investigation results from external facilities prior to the current encounter (see *Histroy of Present Illness* in Figure 9). This configuration ensures that the AI clinician initiates reasoning from a high-fidelity baseline, effectively simulating both primary care consultations and complex referrals that necessitate strategic diagnostic synthesis to reach a definitive diagnosis.

**Specialty:** Nephrology

| | |
|---|---|
| **Basic Information** | **Female, 32 years old, Gardener.** |
| **Chief Complaint** | "I have had sharp, intermittent pain in my left side for two weeks, which became constant and severe two days ago. I've been vomiting, coughing, and I feel like I can't catch my breath. A clinic gave me oxygen, but it's getting worse." |
| **History of Present Illness** | The patient experienced sharp left flank pain 2 weeks ago. Symptoms escalated 2 days prior to admission with persistent pain and nausea. On the day of admission, she developed a dry cough, fever, and dyspnea. Initial evaluation at an outside clinic revealed tachycardia, tachypnea, and hypoxia ($SpO_2$ 89% on room air). Labs showed leukocytosis, hypoalbuminemia (2.8 g/dL), and significant hematuria/proteinuria. Despite oxygen therapy, her dyspnea and lower extremity edema progressed, leading to emergency transfer. |
| **Past Medical History** | Previously healthy. No history of hypertension or diabetes. *Surgical history:* Appendectomy. No known allergies. No recent travel or vaccinations. |
| **Family History** | Parents: Diabetes and hypertension. **Sister:** History of kidney disease, started hemodialysis at age 17 and deceased at 18. No family history of thrombosis or malignancy. |
| **Personal History** | Lives in Massachusetts. No smoking, vaping, or illicit drug use. Occasional alcohol consumption. Nulligravida. |
| **Physical Examination Findings** | **Vital Signs:** T 39.6°C, BP 102/58 mmHg, HR 122 bpm, RR 28 bpm, $SpO_2$ 96% (on 3L $O_2$). BMI 30.1.
**General:** Patient in acute distress.
**Chest:** Coarse crackles heard bilaterally during inspiration and expiration.
**Abdomen:** Tenderness localized to the left lower quadrant.
**Extremities:** No peripheral edema noted at the time of exam; joints normal. |
| **Diagnostic Results** | <ul><li>**Laboratory:** WBC 19,000/$\mu$L (↑); Albumin 2.8 g/dL (↓); $CO_2$ 23 mmol/L (↓); AG 18 mmol/L; HbA1c 6.2% (↑).</li><li>**Urinalysis:** Blood 3+, Protein 3+; RBC ¿100/HPF, WBC 10-20/HPF.</li><li>**Quantification:** Urine protein/creatinine ratio 3.5; Protein excretion ¿4 g/day.</li><li>**Serology: Anti-PLA2R Antibody positive** (ELISA: 400.3 RU/mL, Ref ¡14).</li><li>**Imaging (CXR/CT):** Right lower lobe pulmonary embolism (PE); Bilateral renal vein thrombosis (Left: nearly occlusive; Right: non-occlusive). Multifocal peribronchial consolidations and ground-glass opacities.</li></ul> |
| **Gold-Standard Diagnosis** | PLA2R-associated Membranous Nephropathy. |

*Figure 9.* A representative organized patient record from **CompDiag-Bench**, demonstrating the granularity of clinical findings.

## C. Baseline Model List

**GPT-5.2 (Singh et al., 2025)** A general-purpose model designed for broad task coverage, delivering consistent performance gains across mathematics, coding, science, and tool-calling workloads, with enhanced long-form coherence and more reliable tool use.

**Gemini 3 Flash (Comanici et al., 2025)** A high speed, high value thinking model designed for agentic workflows, multi round chat, and coding assistance.

**Claude Sonnet 4.5 (Anthropic, 2025)** A model optimized for real-world agents and coding workflows, designed for extended autonomous operation, maintaining task continuity across sessions and providing fact-based progress tracking.

**GPT-5 mini (Singh et al., 2025)** A compact version of GPT-5, designed to handle lighter-weight reasoning tasks.

**GPT-4.1 (Achiam et al., 2023)** A model optimized for advanced instruction following, real-world software engineering, and long-context reasoning.

**DeepSeek-V3.2 (DeepSeek-AI et al., 2025)** A model designed to harmonize high computational efficiency with strong reasoning and agentic tool-use performance, which enables thinking by default.

**DeepSeek-V3.1 (DeepSeek-AI, 2024)** A large hybrid reasoning model (671B parameters, 37B active) that supports both thinking and non-thinking modes via prompt templates. All references to DeepSeek-V3.1 above pertain to the `DeepSeek-V3.1-Terminus` edition which enables thinking.

**GLM-4.7 (Team et al., 2025a)** A model featuring upgrades in two key areas: enhanced programming capabilities and more stable multi-step reasoning/execution, which enables thinking by default.

**Qwen3-32B (Yang et al., 2025)** A dense 32.8B parameter causal language model from the Qwen3 series, optimized for both complex reasoning and efficient dialogue.

**Qwen2.5-72B-Instruct (Qwen et al., 2025)** A 72.7B parameter model from the Qwen2.5 series, optimized for expanded knowledge, stronger coding and mathematical reasoning, improved long-context and structured data handling, more reliable structured output generation, and greater robustness to diverse system prompts.

**Qwen3-8B (Yang et al., 2025)** A dense 8.2B parameter causal language model from the Qwen3 series, designed for both reasoning-heavy tasks and efficient dialogue.

**GLM-Z1-Flash (GLM et al., 2024)** A free-to-use reasoning model.

**Qwen2.5-7B-Instruct (Qwen et al., 2025)** A 7.61B parameter model from the Qwen2.5 series.

**Qwen2.5-3B-Instruct (Qwen et al., 2025)** A 3.09B parameter model from the Qwen2.5 series.

**Baichuan-M2-32B (Team et al., 2025b)** A medical-specific model designed for real-world medical reasoning tasks, built upon Qwen2.5-32B with an innovative Large Verifier System.

**MedGemma-27B (Sellergren et al., 2025)** A medical-specific 27B parameter model optimized for inference-time computation on medical reasoning.

**Baichuan-M3 (Team, 2025)** A medical-specific 235B parameter model trained to explicitly model the clinical decision-making process, with the aim of improving usability and reliability in real-world medical practice.

## D. Training Details

### D.1. Training Data

To instill robust clinical reasoning and strategic rigor in `Salus`, we curated a large-scale training corpus derived from 8,645 high-quality patient records.

**Teacher-Guided SFT Data Generation.** We employed `DeepSeek-V3.1` as a teacher model to perform multi-round diagnostic rollouts on the training cases following the pipeline illustrated in Figure 3c. During each interaction, the teacher model sequentially executes the roles of the DR, SC, and WP. To ensure the strategic quality of the supervision, we implement a **Logic-Consistency Filter** specifically for the Strategic Controller (SC).

If the teacher model issues a "Stop" signal ($s_t = $ Stop) while the primary diagnosis $d^{(1)}$ is incorrect, we trigger up to two additional regeneration attempts. If the model persistently opts for premature closure after three total attempts, the corresponding $(X_{SC}, Y_{SC})$ pair for the SC is discarded to prevent the student model from learning suboptimal termination logic. Crucially, to maximize the utility of the clinical case, we manually override the signal to "Continue," allowing the trajectory rollout to proceed and generate subsequent high-quality diagnostic evidence. Through this filtering process, we compiled a finalized SFT dataset comprising 28,928 samples for the DR, 25,483 for the SC, and 21,027 for the WP.

**Data Statistics and Preprocessing.** The finalized corpus exhibits a diverse distribution of clinical complexity and dialogue depth. The average dialogue length is 2.63 rounds, with a minimum of 1 and a maximum of 8 rounds. In terms of token density, the average sample length is 2,574.25 tokens (min: 168; max: 11,029). To accommodate computational constraints and maintain focus on the most relevant clinical context, all sequences were truncated to a maximum length of 6,656 tokens.

**Reinforcement Learning Data.** For the RL stage, we utilize the same case distribution as the SFT phase. Unlike the SFT stage, which relies on the teacher model's responses ($Y_i$), the RL stage only utilizes the initial clinical prompts and the sampled interaction states ($X_i, i \in \{$DR, SC$\}$). This strategy effectively decouples the policy's exploration from the latency-heavy environment feedback loop. Since the *Diagnostic Environment* is itself powered by a large-scale LLM, synchronous multi-round interactions during RL training would introduce prohibitive computational overhead and inference latency. By operating on pre-sampled interaction states, we eliminate the need for real-time test result generation by the environment LLM. This transforms the complex sequential diagnostic trajectory into a resource-efficient training process, allowing for extensive policy exploration without the expensive financial costs associated with continuous environment-agent synchronization.

### D.2. Train-Test State Distribution

Since RL is performed on sampled cumulative clinical histories $H_t$, we examine whether the training states cover the state distribution encountered at test time. We characterize each state by the number of accumulated test results and the token length of the history. As shown in Table 16, the test states are largely contained within the training-state support, with coverage rates of $100\%$ for #Tests and $96.6\%$ for #Tokens. This analysis indicates that the offline RL setup does not introduce a severe train-test mismatch in the observed state space.

*Table 16.* Distribution of cumulative clinical history states $H_t$ in training and testing. Coverage is measured as the fraction of test states falling within the corresponding training-state range.

| Dataset | Metric | Mean | Std | Median | Min | Max | Coverage |
|---------|--------|------|-----|--------|-----|-----|----------|
| Train | #Tests | 8.6 | 9.5 | 6 | 0 | 63 | – |
| Test | #Tests | 11.0 | 11.7 | 8 | 0 | 60 | 100% |
| Train | #Tokens | 1628.6 | 540.8 | 1540 | 429 | 4724 | – |
| Test | #Tokens | 1460.4 | 808.2 | 1284 | 279 | 5543 | 96.6% |

### D.3. Training Implementation and Hyper-parameters

The training of `Salus-7B` is conducted in two stages. The **Supervised Fine-Tuning (SFT)** phase is executed for 3 epochs on 4 NVIDIA H800 GPUs using `TRL` framework (von Werra et al., 2020). We utilize a batch size of 64 and a peak learning rate of $2 \times 10^{-5}$, governed by a cosine learning rate scheduler with a 0.1 warmup ratio. To accommodate long-form Chain-of-Thought (CoT) reasoning, the maximum generation length is set to 6,656 tokens.

The **Reinforcement Learning (RL)** stage is implemented using the `rllm` framework (Tan et al., 2025) across 16 NVIDIA H800 GPUs. The joint optimization of the DR and SC components spans 500 training steps (reduced to 250 steps for ablation studies). To monitor convergence and prevent overfitting, we reserve a random 1% split of the training data as a validation set; the final model checkpoint is selected based on the highest Top-1 diagnostic accuracy on this held-out set. The KL divergence coefficient $\beta$ is set to 0.001 to maintain policy stability. To capture more extensive strategic deliberations, the maximum output length is extended to 8,192 tokens during this phase.

**Training Time and API Cost.** The SFT phase takes approximately 8 hours, and the RL phase takes approximately 45.5 hours. The dominant API costs come from teacher trajectory generation and LLM-as-a-Judge reward computation. In our experiments, teacher trajectory generation costs approximately $800, while judge calls during RL cost approximately $1,000.

## E. LLM Prompts

---

### Prompt of the DR

You are a medical expert. Your task is to reason through the top 5 differential diagnoses based on the provided patient medical records.

**Core Requirements:**
- **Diagnostic Ranking**: List diagnoses in order of probability, from highest to lowest.
- **Diagnostic Completeness**: Each diagnosis must be complete and specific (e.g., use "Right lower lobe pneumonia" instead of "Pneumonia"; "Coronary heart disease, unstable angina" instead of "Heart disease"). It may include the primary disease and related complications/comorbidities (e.g., "Type 2 diabetes mellitus with community-acquired pneumonia").
- **Diagnostic Competition**: The diagnoses in the list should be competing alternatives (i.e., differential diagnoses). **Do not split different aspects of a single pathological process into separate entries** (e.g., do not list "Community-acquired pneumonia" and "Fever" as two separate diagnoses).
- **Focus on Diagnosis**: Your response should focus on the diagnostic reasoning process and the final list of diagnoses. **Strictly prohibit** providing any treatment plans, medication suggestions, or health guidance. Do not include raw examination results in the final diagnosis list.
- **Number of Diagnoses**: You may provide fewer than 5 differential diagnoses if appropriate.

The following is the patient's information:
<Medical Record>
{}
</Medical Record>

Output Format:
Step-by-step analysis...
<answer>
Diagnosis 1
Diagnosis 2
...
</answer>

Output Example:
... (Step-by-step analysis)
<answer>
Tuberculous meningitis/meningoencephalitis, with community-acquired pneumonia
Paranasal abscess, complicated by systemic tuberculous infection
</answer>

Now, please provide a step-by-step analysis first, and then output several competing and complete diagnostic options within the <answer> tags. Do not provide any other content.

---

## Prompt of the SC

You are a medical expert responsible for advising a clinician on whether to proceed with further auxiliary examinations or provide a final diagnosis, based on the patient's existing medical records and a list of preliminary differential diagnoses.

**Conditions for providing a final diagnosis directly (at least one must be met):**
1. **Unique Diagnosis with Conclusive Evidence**: There is only one disease in the differential diagnosis list that fully explains the patient's core symptoms, and there is sufficient evidence (e.g., "gold standard" test results) to support it.
2. **Clear Diagnostic Hierarchy and Causality**: One diagnosis in the list is the root cause (primary diagnosis), and all other diagnoses can be clearly explained as the **direct triggers, complications, or associated manifestations** of that diagnosis, rather than independent competing diagnoses. Meanwhile, the primary diagnosis must be supported by sufficient evidence.
3. **Principle of Preponderance of Evidence**: One specific diagnosis has a complete and logically rigorous chain of evidence that perfectly fits all information in the medical record (including symptoms, signs, and existing tests) without any contradictions, and is supported by sufficient evidence; whereas all other diagnoses in the list have key points of doubt that cannot explain the current medical record.

**Note:** A final diagnosis can only be given if there is sufficient evidence to support it; otherwise, you must proceed with further auxiliary examinations.

If you choose to provide a final diagnosis, you must output the top-1 diagnosis.

**Output Format Requirements:**
- The final result must only contain the decision to continue examinations or the final full diagnosis: **Strictly Prohibited** from providing any treatment plans, medication suggestions, or health guidance. Do not output non-contradictory information such as existing test results.
- If you choose to continue testing, output: `Proceed with further auxiliary examinations`.
- If you choose to provide a final diagnosis, output the core diagnosis in the following format: `Your diagnosis is: xxx (Core Diagnosis)`.
- Provide a step-by-step analysis first, then place the final result within `<answer>` tags.

**Output Example 1**
Step-by-step analysis...
<answer>
Proceed with further auxiliary examinations
</answer>

**Output Example 2**
... (Step-by-step analysis)
<answer>
Your diagnosis is: Cerebral Venous Sinus Thrombosis.
</answer>

**Existing Patient Medical Record:**
<Medical Record>
{}
</Medical Record>

**Preliminary Differential Diagnosis List:**
<Diagnosis>
{}
</Diagnosis>

Please analyze step-by-step to determine whether the clinician should proceed with further auxiliary examinations or provide a final diagnosis.

## Prompt of the WP

You are a medical expert responsible for providing suggestions for next-step diagnostic tests to clinicians based on the **Patient's Existing Medical Record** and the **Preliminary Differential Diagnosis List**.

### Core Principles for Decision Making
Please follow the "stepwise testing" strategy of clinical medicine:
1. **Necessity First**: Only recommend tests that can significantly alter clinical decision-making or the direction of the differential diagnosis.
2. **No Redundant Tests**: Generally, do not recommend tests that have already been performed.
3. **Precision and Cost Control**:
   - Prioritize non-invasive, low-cost, and high-sensitivity screening items.
   - Expensive or invasive tests (e.g., biopsy, PET-CT, genetic testing) should only be recommended when low-cost tests cannot confirm the diagnosis and the test is crucial for distinguishing between core differential diagnoses.
4. **Specific Naming**: The names of examination items must be sufficiently specific. For example, for CT/MRI, the body part and method must be specified (e.g., Non-contrast Head CT, Contrast-enhanced Abdominal CT, Coronary CTA).

### Testing Goals
The recommended combination of tests should aim to achieve one of the following goals:
- **Confirmation**: Obtain "gold standard" evidence.
- **Exclusion**: Safely rule out high-risk differential diagnoses through key negative results.
- **Differentiation**: Effectively distinguish between the two most similar competing diagnoses in the list.

### Output Format Requirements
- Only output recommended examination items: Your response should focus on the recommended tests. **Strictly prohibit** providing any treatment plans, medication suggestions, or health guidance. Do not output non-contradictory information regarding patient results; do not output the analysis process within the list.
- Examination items must be sufficiently specific and definite. Strictly prohibit the use of vague expressions like "suggest considering" or "or". The output list should be a definitive, mandatory clinical order list.
- Output one specific examination item per line. **It is prohibited to place multiple items on the same line** (e.g., expressions like "Liver stiffness measurement or serum liver fibrosis marker test" or "CBC + Coagulation + Liver Function" are prohibited).
- Place the content inside `<answer>` tags.

Output Format Example: The first line inside the `<answer>` tag must be fixed as "Requesting the following supplementary examinations:"
Step-by-step analysis...
<answer>
Requesting the following supplementary examinations:
- Serum liver fibrosis marker test
- Complete blood count
- Fecal occult blood test
</answer>

Patient's existing medical record information:
<MedicalRecord>
{}
</MedicalRecord>

Preliminary differential diagnosis list:
<Diagnosis>
{}
</Diagnosis>

Please provide a step-by-step analysis based on the medical record and diagnosis first, and then recommend several supplementary examinations inside the `<answer>` tag according to the format.

## Prompt of the Diagnostic Environment

You are a senior clinical medical expert. Your task is to **output the results of corresponding auxiliary examinations based on the provided medical record**.

## Processing Rules
1. **Determine if the examination results are explicitly recorded in the medical record**
  - If the **complete results** of the examination are **explicitly recorded** in the medical record, output the content exactly as it appears.
  - If the results are **not explicitly recorded**, or if only a portion is recorded while **missing other parts**, you must make reasonable clinical inferences based on other information in the medical record to provide results that align with clinical reality.
    - For numerical results: first infer a reasonable range, then sample a specific value within that range.
    - All numerical results must be **labeled to indicate whether they are abnormal**, for example, using "↑" for elevated, "↓" for decreased, and "(normal)" if the value is within the reference range.

2. **Prohibit revealing any diagnostic information in the results**, such as "consistent with [Disease X]" or "inferred based on [Disease X]". The output results should simulate an actual report obtained from a real examination, maintaining objectivity and neutrality; the final result should not appear to be fabricated.

3. **Focus on Readability**: If there are uncommon English abbreviations, provide the full Chinese name. For example:
  - If the medical record says: "Pancreas: 1mm low-density lesion in the head, consistent with previous IPMN."
  - You should output: "Pancreas: 1mm low-density lesion in the head, consistent with previous IPMN (Intraductal Papillary Mucinous Neoplasm)."

4. **Abnormalities First**: To help doctors quickly read useful information.
  - If an examination includes multiple indicators/aspects, **prioritize describing abnormal findings**, followed by normal findings or those with no significant abnormalities.
  - If there are no abnormalities at all, start with the phrase "No abnormalities found," followed by the description of specific results.

## Output Format
Provide a step-by-step analysis first, then output the results within XML tags. Place the "hit" status in `<hit>` and the examination results in `<answer>`. Specifically:
[Step-by-step analysis]
<hit>Enter "yes" if at least one examination item is explicitly or partially recorded in the medical record; otherwise, enter "no". No other text is allowed in this tag.</hit>
<answer>
Examination Name (must match the provided name): Examination Results (all results for that examination placed in a single line)
</answer>

**Output Example 1**: Assuming the requested examination is `Urinalysis and Urine Culture`, and the medical record has no record of this.
... (Step-by-step analysis)
<hit>no</hit>
<answer>
Urinalysis and Urine Culture: Glucose 3+ ↑, Ketones 1+ ↑, Escherichia coli 1.2×10^5 CFU/mL ↑ (Ref <10^4 CFU/mL); Urine protein negative, RBC and WBC in urine sediment are normal.
</answer>

**Output Example 2**: Assuming the requested examination is `Abdominal Ultrasound`, and the medical record has no record of this.
... (Step-by-step analysis)
<hit>no</hit>
<answer>
Abdominal Ultrasound: No abnormalities found. Liver, gallbladder, pancreas, spleen, and both kidneys are normal in size and morphology; no space-occupying lesions detected.
</answer>

The Reference Medical Record:
<record>
{}
</record>

The Request of auxiliary examinations:
{}

Now let's think step by step, and finally output the results with <answer> tags.

## Prompt of four-level rubric for diagnosis evaluation

You are a medical expert responsible for grading: based on the provided standard answers, you will perform a precise and objective evaluation of a clinician's diagnosis. You must strictly follow the **Assessment Process** below to provide your judgment and explanation.

# Assessment Process
You must strictly adhere to the following two steps for evaluation:

## Step 1: Core Diagnosis Deconstruction
Before evaluating, you must first analyze the [Final Diagnosis (Ground Truth)] and break it down into:
1. **Core Diagnosis**:
   * The primary reason for the patient's current visit, consisting of one or more serious conditions within the diagnosis.
   * Includes, but is not limited to, life-threatening conditions or those requiring urgent medical intervention.
   * *Evaluation Principle*: This serves as the primary basis for the subsequent assessment.
2. **Secondary Diagnosis**:
   * Can be long-term stable chronic underlying conditions (e.g., well-controlled hypertension) that are generally not life-threatening and do not require urgent intervention.
   * Can also be concomitant diseases or symptoms directly caused by the core diagnosis.
   * *Evaluation Principle*: These are usually ignored during assessment. If the doctor omits a secondary diagnosis, **no** points are deducted.

## Step 2: Four-Level Evaluation Based on Treatment Consequences
Based on the "Core Diagnosis," compare it with the "Doctor's Diagnosis." Choose one of the following four levels based on the **impact on the primary treatment plan for the core diagnosis**:

### 1. Completely Correct
* **Definition**: The doctor's diagnosis accurately covers all **core diagnoses**.
* **Characteristics**:
   * The etiology, location, and nature are completely consistent.
   * Only secondary diagnoses are omitted.

### 2. Clinically Acceptable
* **Definition**: The doctor's diagnosis is correct in its **core direction**, with flaws only in **precision**.
* **Key Characteristics**:
   * **Precision Flaws**: Used a correct hypernym (e.g., wrote "Bacterial Pneumonia" instead of "Streptococcus Pneumoniae Pneumonia"), or imprecise typing/grading.
* **Gold Standard for Judgment**: **The subsequent treatment plan would not differ significantly.** A prescription issued by a doctor based on this diagnosis would cover the core treatment required by the patient (e.g., correct antibiotic coverage, basically consistent surgical approach).

### 3. Partially Correct
* **Definition**: The general direction of the doctor's diagnosis (e.g., system or symptoms) makes some sense, but there are **critical omissions or vagueness**, leading to significant defects in the treatment plan.
* **Key Characteristics**:
   * **Core Omission**: Among multiple co-existing core diagnoses, one was missed.
   * **Vague Precision**: The diagnosis is too general, making it impossible to perform necessary specific treatments (e.g., "Septic Shock" without specifying the source of infection, leading to antibiotic misuse or ineffectiveness); or only diagnosing a severe syndrome caused by the core diagnosis (e.g., shock, heart failure) without diagnosing the primary disease.
   * **Typing Error Leading to Treatment Change**: For example, misjudging a type requiring surgery as a type for conservative treatment.
* **Gold Standard for Judgment**: **There is a significant difference in the treatment plan**, or a **core diagnosis is missing**. Although the doctor did not completely misdiagnose (it **makes some sense**), the patient cannot receive targeted key treatment (e.g., missed thrombolytics, missed a specific surgery).

### 4. Incorrect
* **Definition**: A **completely wrong** diagnosis that neither covers any core diagnosis nor has any reference value.
* **Characteristics**: Misdiagnosing Disease A as Disease B (entirely different mechanisms), or only diagnosing "Abdominal pain" or "Pending investigation."
* **Gold Standard for Judgment**: **Treatment is completely wrong or delayed.**

# Notes
1. Secondary diagnoses are subordinate compared to core diagnoses; a secondary diagnosis is not tied to a specific disease. For example, "Fatty liver" itself can be a core diagnosis, but if the patient also has "Lung cancer," then "Lung cancer" becomes the core diagnosis and "Fatty liver" becomes a secondary diagnosis.
2. **The treatment plan only considers the general diagnostic pathway for the core diagnosis**; it does not consider secondary diagnoses, nor does it require detailed or specific treatment regimens.
3. Compared to "Completely Correct," if there is an error in precision, it may be categorized as "Clinically Acceptable" or "Partially Correct." The specific category depends on whether the degree of error is acceptable based on the subsequent treatment plan.

# Response Format
Step-by-step analysis process
<answer>Judgment Label</answer>

The judgment label must be one of the 4 categories, for example:
<answer>Clinically Acceptable</answer>

Final Diagnosis (Ground Truth): {}
Doctor's Diagnosis (To be evaluated): {}

Please evaluate and classify the clinician's diagnosis based on the provided medical information. Provide the step-by-step analysis first, and place the final judgment within the <answer></answer> tags.

