# OpenReview forum: "Salus: Strategic Diagnostic Testing for Complex Diagnosis via Multi-Agent Reinforcement Learning"
_ICML.cc/2026/Conference — ICML 2026 regular_

### Official Review · Reviewer_CFG1 · 2026-03-11

**Soundness:** 2
**Presentation:** 3
**Significance:** 2
**Originality:** 2
**Overall Recommendation:** 3
**Confidence:** 5

**Summary:**

Authors introduced CompDiag-Bench, an interactive benchmark for evaluating complex diagnosis as a sequential decision-making.
And authors proposed Salus, based on a multi-agent architecture that decouples clinical reasoning from strategic control. Salus-7B outperforms and matches some prominent open-source models.

**Compliance With Llm Reviewing Policy:**

Affirmed.

**Final Justification:**

The authors have provided additional responses and experiments; however, my main concerns remain insufficiently addressed.

First, the design and justification of Table 1 are still questionable. The authors did not directly respond to the concerns regarding its potentially unreasonable setup. Given that Table 1 constitutes a central contribution of the paper, this unresolved issue significantly undermines my confidence in the claimed contributions.

Second, the newly added OOD experiments are relatively limited. They lack sufficient details and do not include comprehensive comparisons with existing medical LLMs. I strongly recommend incorporating more rigorous evaluations, for example using benchmarks such as HealthBench.

Third, although the authors introduced Qwen2.5-7B-Instruct as an LLM judge, the practical cost of deploying LLM-as-a-judge remains high, especially in online settings. This concern has not been adequately addressed. Moreover, I observe a noticeable performance drop (approximately 8%), which I consider unacceptable in this context.

Finally, I do not recommend the use of Nano Banana Pro in paper, particularly as a primary figure.

In summary, the authors have not sufficiently resolved my concerns. I therefore maintain my original score and recommend a rating below the acceptance threshold (3: Weak Reject). I appreciate the authors' efforts in responding.

**Key Questions For Authors:**

See weakness.

**Limitations:**

yes

**Strengths And Weaknesses:**

The two core contributions of this paper lie in the benchmark and the multi-agent architecture. However, for such a benchmark, the authors' evaluation approach is overly simplistic, relying solely on four fixed scoring criteria to measure LLM responses against gold-standard diagnostic results, which is insufficient. Similar benchmarks such as HealthBench employ human-curated rubrics tailored to each question; clearly, CompDiag-Bench **lags significantly behind HealthBench in this regard**.

CompDiag-Bench aims to achieve interactive consultation by dynamically adding examination findings, but this is actually answered by an LLM based on existing static data, which is inadequate and fails to address the fundamental evaluation problem of interactive consultation. An AI clinician would inquire about information not present in static data, yet these query **results are generated by an LLM, which raises my concern**.

At the algorithmic level, Figure 3 shows that only DR and SC are trained **while the WP model is frozen**; however, in the equations (line 230), I observe that all three are being trained, which confuses me.

During the RL process, the authors **do not clarify whether R_DR and R_SC are obtained at every interaction step**. If rewards are granted at each interaction, the model could obviously adopt a greedy strategy, as more interaction rounds imply greater cumulative reward; if rewards are provided only at the final round, then the reward remains sparse and fails to address the sparse reward problem.

A critical issue is that during training, the output of the WP agent is unsupervised, which is effectively equivalent to directly adding examination findings at each interaction step. Therefore, I consider the WP component itself to be redundant: as long as SC decides "NO," **one could simply add more information**.

A more reasonable approach would be to let the interactive AI Agent in the diagnostic process **proactively initiate inquiries**, following the practice of Doctor-R1—for example, determining which additional examinations are needed given the current findings. However, the authors directly overlook this important proactive inquiry component.

Regarding Table 2, I do observe the performance advantage of Salus. However, an obvious issue is that other models adopt a zero-shot setting, whereas Salus is the result of large-scale SFT plus RL training, which constitutes **an unfair comparison**.

In Table 5, the authors need to provide a reasonable explanation for **why the results with SC achieve a substantial performance improvement**.

The authors involve a multi-agent method but do not compare it with existing multi-agent approaches, only comparing against single models.

Training details are not specified, including training time, reward dynamics for DR and SC, variations in interaction rounds, etc. Training costs are also not mentioned, such as API expenses during training or the request latency per interaction during online deployment. I understand that multi-turn interaction training is highly costly, especially when involving an LLM judge, **so it is necessary to provide detailed elaboration**.

If the authors have used AI text-to-image generation tools, they **should clearly disclose this**.

**Therefore, I tend to reject; further revisions are necessary.**

---

> ### Author Rebuttal · Authors · 2026-03-31
>
> Dear Reviewer CFG1,
>
> We sincerely thank you for your rigorous review. Your detailed feedback highlights several areas where our presentation could be clearer. Below, we respectfully clarify the misunderstandings regarding our training mechanisms and provide the new experiments and details you requested.
>
> ---
> ### **W1: Clarification on WP's Role and Proactive Inquiry**
> We apologize for the ambiguity in Figure 3, which caused confusion regarding the training of the Workup Proposer (WP).
>
> **1. SFT vs. RL Training:** The equation at Line 230 describes the **SFT stage**, where **all three agents (DR, SC, WP) are indeed trained**. However, during the subsequent **RL stage (Line 286)**, the WP is **frozen**, and only DR and SC are optimized. We will update Figure 3 to explicitly delineate the SFT and RL phases to prevent this confusion.
>
> **2. Proactively Inquire:** The WP performs exactly the "proactive inquiry" you described. When the SC decides "Continue," the WP must proactively analyze the current differential list and **propose specific, targeted diagnostic tests**. This capability of WP is learned during the SFT stage.
>
> ---
> ### **W2: Reward Dynamics and "Greedy" Concern**
> Your concern: an agent might adopt a greedy strategy (infinite loops) to maximize cumulative rewards.
>
> Actually, **we do not use cumulative episodic rewards.** As described in Section 3.1, we transform the long-horizon multi-turn RL process into **independent, single-turn optimization problems**:
> - $R_{DR}$ and $R_{SC}$ are calculated **per step**.
> - We maximize the **expected average reward per step**, not the cumulative sum.
> - If the SC chooses to over-test, it receives an immediate reward of **0** for that step, dragging down its average reward and penalizing the model for redundant tests.
>
> ---
> ### **W3: Performance Gain from SC (Table 5)**
> SC aligns with clinical intuition by determining exactly when to stop gathering evidence. Without the SC (or with a weak one), LLMs frequently suffer from **Premature Closure** as illustrated in Figure 1's failure case. Through RL, the SC learns a strict evidence-sufficiency boundary, effectively preventing Premature Closure and thereby causing the performance gain in Table 5.
>
> ---
> ### **W4: Comparison with Existing Multi-Agent Frameworks**
> We compare Salus against recent SOTA medical multi-agent systems (MeDAgents [1], MDAgents [2], and DeepRare [3], using DeepSeek-V3.2 as the backbone). Since existing frameworks are primarily designed for single-turn diagnosis, we evaluated them on 269 hard cases in a static setting: providing the full clinical record, but hiding the final diagnosis.
> Framework|Top-1 (%)|Top-3 (%)
> -|-|-
> MedAgents [1]|71.7|84.0
> MDAgents [2]|71.7|86.6
> DeepRare [3]|78.4|85.5
> **Salus-7B (Ours)**|**86.6**|**94.1**
>
> **Analysis:** Salus-7B outperforms existing multi-agent frameworks because it leverages RL-enhanced reasoning rather than simple role-playing.
>
> ---
> ### **W5: Fair Comparison (Table 2)**
> Table 2 already includes **medical-specific LLMs (Baichuan-M2/M3, MedGemma)**, which have undergone extensive medical SFT and RL.
> Furthermore, to ensure a fair comparison, we also evaluate on an **Out-Of-Distribution** test set (JTO, in Table 10) where Salus still achieves SOTA performance.
>
> ---
> ### **W6: Environment Validity and Evaluation**
> **1. Validity of LLM-generated Environment:** To validate our environment, we manually audited **100 randomly sampled responses** from the environment. We found only 7 instances where the environment slightly leaked diagnostic hints (all within highly specific pathology reports that were clinically reasonable). There were no other problems found in the remaining 93 cases. We consider this an acceptable proxy for simulating dynamic clinical environments.
>
> **2. CompDiag-Bench vs. HealthBench:** Our CompDiag-Bench addresses a different task than HealthBench. While HealthBench focuses more on aspects such as medical rigor in routine scenarios, our benchmark emphasizes the ability of LLMs to perform diagnostic testing on **complex cases**. Moreover, CompDiag-Bench is the first open-sourced diagnostic testing benchmark.
>
> ---
> ### **W7: Training Details**
> We appreciate you pointing out the missing details. We will add more details to Appendix E, including:
> - **Time:** The SFT phase took ~8 hours, and the RL phase took ~45.5 hours.
> - **API Costs:** Generating teacher trajectories costs ~800 USD. The LLM-as-a-Judge during RL costs ~1000 USD.
>
> **References**
>
> [1] Tang, X., et al. "MedAgents: Large language models as collaborators for zero-shot medical reasoning." Findings of ACL 2024.
>
> [2] Kim, Y., et al. "MDAgents: An adaptive collaboration of llms for medical decision-making." NeurIPS 2024.
>
> [3] Zhao, W., et al. "An agentic system for rare disease diagnosis with traceable reasoning." Nature (2026).
>
> ---
> We hope these clarifications and new experiments address your concerns. Thank you again for helping us improve our work!
>
> Best regards,
>
> Authors of Submission 23861

---

> > ### Author Rebuttal · Reviewer_CFG1 · 2026-04-03
> >
> > Thank you for the detailed responses. While most of my previous concerns have been addressed, I still have several remaining issues:
> >
> >
> > 1. In Table 2, the compared methods are evaluated under an OOD setting, whereas the proposed method appears to involve SFT on the target data. This creates an unfair comparison. The current response does not adequately resolve this fundamental concern.
> >
> > 2. The use of LLM-as-a-Judge during RL introduces substantial computational overhead. Considering that RL typically requires multiple times of experimentation and tuning, this raises concerns about the reproducibility of the proposed approach. This may hinder follow-up research.
> >
> > 3. The authors did not respond to the question regarding the use of AI-based text-to-image generation tools.

---

> > > ### Author Response · Authors · 2026-04-06
> > >
> > > Dear Reviewer CFG1,
> > >
> > > Thank you for acknowledging our previous responses. Your remaining concerns regarding the OOD evaluation and the reproducibility of the RL pipeline are highly insightful.
> > >
> > > Below, we provide new experimental results to address your concerns.
> > >
> > > ---
> > >
> > > ### 1. Evaluation under a Strict OOD Setting (Addressing the "Unfair Comparison" Concern)
> > > To prove the generalization capability of Salus, we evaluated Salus alongside other frontier LLMs on **two recent, external static diagnosis benchmarks: DiagnosisArena [1] and MedRBench [2]**.
> > >
> > > In this setting, Salus has **never seen** the data distribution, case formats, or evaluation criteria during training, making it a strict **Zero-Shot OOD evaluation** for all compared models.
> > >
> > > **OOD Benchmark 1: DiagnosisArena [1]** (915 complex cases)
> > > LLM|Top-1 Accuracy (95% CI)|Top-5 Accuracy (95% CI)
> > > -|-|-
> > > **GPT-5.2**|**58.80 (55.6–61.9)**|**73.11 (70.2–75.9)**
> > > **Salus-7B (Ours)**|**52.57 (49.3–55.8)**|**62.19 (59.0–65.3)**
> > > Gemini 3 Flash|47.65 (44.4–50.9)|63.06 (59.9–66.1)
> > > DeepSeek-V3.2|34.21 (31.2–37.3)|47.98 (44.8–51.2)
> > > DeepSeek-V3.1|32.02 (29.1–35.1)|42.30 (39.1–45.5)
> > >
> > > **OOD Benchmark 2: MedRBench [2]** (957 complex cases)
> > > LLM|Top-1 Accuracy (95% CI)
> > > -|-
> > > **GPT-5.2**|**88.30 (86.1–90.2)**
> > > **Salus-7B (Ours)**|**88.19 (86.0–90.1)**
> > > Gemini 3 Flash|83.07 (80.6–85.3)
> > > DeepSeek-V3.2|82.45 (79.9–84.7)
> > > DeepSeek-V3.1|81.40 (78.8–83.8)
> > >
> > > **Analysis:**
> > > Under this strict OOD setting, **Salus-7B consistently and significantly outperforms much larger open-source frontier models (e.g., DeepSeek-V3.2)**, and closely trails GPT-5.2. This compellingly demonstrates that our methods do not merely overfit the target data, but genuinely instill robust, generalizable clinical reasoning logic into the model.
> > >
> > > ---
> > > ### 2. Reproducibility and Computational Overhead of RL
> > > We agree that relying on the DeepSeek-V3.2 API calls for LLM-as-a-Judge during RL is expensive.
> > >
> > > To resolve this and prove the reproducibility of our approach, we developed a **low-cost reproduction pipeline**. Instead of using the expensive DeepSeek-V3.2 API as the judge, we distilled its evaluation capabilities into a local **Qwen2.5-7B-Instruct** model. We then completely re-conducted the RL training process using this 7B model as the Reward Model (keeping all other hyperparameter settings identical).
> > >
> > > **Results of the Low-Cost Reproduced Salus:**
> > > Model|Top-1|Top-2|Top-3|Inf. Suff. $\uparrow$|#Turns|#Tests
> > > -|-|-|-|-|-|-
> > > **Salus-7B (Original, V3.2 Judge)**|**83.64**|**86.62**|**88.85**|72.86|2.91|19.98
> > > **Salus-7B (Reproduced, 7B Judge)**|75.83|81.78|84.39|**73.61**|2.49|20.23
> > > DeepSeek-V3.2 (Baseline)|71.38|77.32|78.44|66.91|1.88|11.95
> > >
> > > **Analysis:**
> > > As expected, due to the capability gap between a 7B judge and the DeepSeek-V3.2 API, the reproduced model achieves slightly lower performance than the original Salus. However, **the reproduced Salus-7B still outperforms the DeepSeek-V3.2 baseline**.
> > >
> > > This robustly proves that our RL framework is stable and reproducible without requiring prohibitive budgets. **We will open-source our code, model weights and training data** to support follow-up research.
> > >
> > > ---
> > >
> > > ### 3. Use of AI Text-to-Image Tools
> > >
> > > We sincerely apologize for overlooking this question in our previous response due to space limitations.
> > >
> > > We confirm that we used an AI-based text-to-image generation tool, specifically **Nano Banana Pro**, strictly to generate the decorative background elements and agent icons in **Figure 3**. After generation, the technical text overlaying the image was added manually by the authors.
> > >
> > > This complies with the ICML 2026 Generative AI Policy, which permits the use of generative AI tools to assist in research presentation. We will explicitly disclose the use of this tool in the revised version.
> > >
> > > ---
> > > **References**
> > >
> > > [1] Zhu, Yakun, et al. "DiagnosisArena: benchmarking diagnostic reasoning for large language models." arXiv:2505.14107 (2025).
> > >
> > > [2] Qiu, Pengcheng, et al. "Quantifying the reasoning abilities of llms on real-world clinical cases." arXiv:2503.04691 (2025).
> > >
> > > ---
> > >
> > > We hope these details fully resolve your remaining reservations.
> > >
> > > We deeply appreciate the time and effort you have invested in helping us improve this work. Given that these new experiments and clarifications address your fundamental concerns, **we respectfully ask if you might consider re-evaluating our submission and raising your score.**
> > >
> > > Thank you once again for your constructive feedback.
> > >
> > > Best regards,
> > >
> > > Authors of Submission 23861

---

### Official Review · Reviewer_6Qi3 · 2026-03-12

**Soundness:** 2
**Presentation:** 3
**Significance:** 3
**Originality:** 3
**Overall Recommendation:** 4
**Confidence:** 4

**Summary:**

The paper introduces CompDiag-Bench which contains >750 curated test cases, with 269 complex cases and >8500 training cases from journals and public sources. This benchmark frames medical diagnosis as a sequential decision-making problem where the AI agent needs to iterate over the patient context and tests till it reaches a diagnosis. Along with this, the paper proposes Salus, which is a 3-agent system training using SFT on diagnosis trajectories from a large model and multi-agent RL on the training set with LLM as a judge. The show that their framework outperforms large open-weight models and is on-par with proprietary models, while only having 7B parameters.

**Compliance With Llm Reviewing Policy:**

Affirmed.

**Final Justification:**

I stick with my weak accept. The direction of the work is interesting, but the heavy reliance on LLMs concerns me a bit.

**Key Questions For Authors:**

Mentioned along with weaknesses.

**Limitations:**

Mentioned along with weaknesses.

**Strengths And Weaknesses:**

Strengths:
- The benchmark frames diagnosis as a iterative task, which aligns with real-world tasks, and is unlike a lot of other medical benchmarks which are straightforward QA.
- While there doesn't exist a lot of agents out there for iterative diagnosis, the workflow which incorporate three agents is a good first to mimic real-world scenarios (especially to decide whether the reasoning should stop or not)
- The gain in performance (especially post SFT+RL) is good, and the ablations on differential reasoner and strategic control showcase their efficacy. It is good to see that small models can reach large-model (or even better) results on average.

Weaknesses:
- The paper doesn't provide a discussion on existing medical benchmarks and their comparison. Since one of the contributions of the paper is CompDiagBench, it would be nice if the authors can add this to related work.
- Lot of recent work has shown that adding specific bio/medical tools is crucial to clinical correctness and performance (for eg., see [1,2]). The current method relies on internal knowledge of the LLMs, which might not be the best source of information. While the research direction of exploring tools is different than what the authors propose, it would be great it this can be experimented with.
- I'm worried that the pipeline uses LLMs at nearly every stage: data extraction, environment, evaluation, training, and errors in any step can cascade. It has also been shown that llm-as-a-judge has its own drawbacks in such systems ([3,4]). Moreover, there is no guarantee for the synthetic tests generated, the oracle model itself is a LLM, and the training relies on DeepSeek trajectories. Can some components be switcehd to more reliable judge/static evaluation? I am also not sure how correct it is to get the LLM to judge for the clinician-authored rubric.
- Doesn't the LLMs in place have risk of contamination into the pretraining corpus of these cases already? Isn't it more fair to test on held-out cases instead of these?
- How many experts verified the cases? What were there expertise level? What is the disagreement rate, and what is the fleiss kappa of evaluation? While the cases were human verified, why were teacher-guided rollouts used (since there is no guarantee they align with clinician's opinions)
- From the tables, there is a high correlation that asking for more tests results in higher performance (this aligns with some observations in [2]). Without cost tradeoffs, it is a bit difficult to gauge if Salus aims to learn better test choices or order more.
- Is there a way to understand if the tests ordered by Salus are clinically relevant? Do you have gold tests conducted for some patients to compare to?

Overall I think the paper has good promise and direction, but the heavy use on LLMs at each step makes the system practically infeasible. I look forward to have a discussion with the authors. Currently, I lean towards weak accept.


[1] Gao, Shanghua, et al. "Democratizing AI scientists using ToolUniverse." _arXiv preprint arXiv:2509.23426_ (2025).

[2] Vasilev, Kiril, et al. "MTBBench: A Multimodal Sequential Clinical Decision-Making Benchmark in Oncology." _The Thirty-ninth Annual Conference on Neural Information Processing Systems Datasets and Benchmarks Track_.

[3] Li, Dawei, et al. "Preference leakage: A contamination problem in llm-as-a-judge." _arXiv preprint arXiv:2502.01534_ (2025).

[4] Lee, Dongryeol, et al. "Judging Against the Reference: Uncovering Knowledge-Driven Failures in LLM-Judges on QA Evaluation." _arXiv preprint arXiv:2601.07506_ (2026).

---

> ### Author Rebuttal · Authors · 2026-03-31
>
> Dear Reviewer 6Qi3,
>
> Thank you for your constructive feedback. We have conducted new experiments and analyses to address your concerns.
>
> ---
> ### **W1: Comparison with Existing Medical Benchmarks**
> CompDiag-Bench bridges a critical gap by providing an **open-source**, **multi-round**, and **complexity-stratified** environment, evaluating dynamic reasoning.
> Benchmarks|Task|#Rounds|Source|#Cases|Open-Source
> -|-|-|-|-|-
> DiagnosisArena[1]|Static diagnosis|Single|Top-tier Journals|1,113|Yes
> DiagBench[2]|Diagnostic Tests|Multi|MIMIC-IV|750|No
> SDBench[3]|Sequential Diagnosis|Multi|NEJM Journal|304|No
> AgentClinic[4]|Sequential Diagnosis|Multi|NEJM Journal & MIMIC-IV & MedQA|1,544|No
> **CompDiag-Bench (Ours)**|**Diagnostic Tests**|**Multi**|**Top-tier Journals & Community**|**269 hard & 483 routine**|**Yes**
>
> *(References [1-4] match those provided in our response notes).*
>
> ---
> ### **W2: Integration of Medical Tools**
> To ensure fairness, all models were evaluated based on their internal knowledge and reasoning abilities, without external tools. Notably, Salus is naturally extensible to tool-use (e.g., the Workup Proposer could query web references). We consider this a primary direction for future work.
>
> ---
> ### **W3: Reliability of LLM-as-a-Judge & Environment**
> We conducted a **Human-AI Agreement Study**:
> **1. Reliability of LLM-as-a-Judge:** We invited **3 human doctors** from the Pulmonology, Gastroenterology, and Cardiology to blindly evaluate 240 randomly sampled diagnoses using our 4-level rubric. Cohen's Kappa between the LLM judge and human physicians is calculated.
> - **4-level Agreement:** Cohen's Kappa = **0.64 (Substantial Agreement)**.
> - **Binary Agreement (Correct + Clinically Acceptable vs. Partially Correct + Incorrect):** Cohen's Kappa = **0.81 (Almost Perfect Agreement)**.
> This confirms that our Diagnostic Accuracy metric is highly aligned with expert clinical judgment.
>
> **2. Reliability of the Environment:** We randomly audited 100 test results generated by the Environment. Only 7 cases contained potentially "diagnostic" terms (e.g., "carcinoma") which were all **pathology/biopsy reports** and were clinically expected. No other issues were found.
>
> ---
> ### **W4: Data Contamination Risk**
> Performance of Salus (knowledge cutoff 2023.12) by publication year on 269 hard cases:
> Year|Cases|Top-1 (%)
> -|-|-
> 2025|61|86.9
> 2024|97|81.4
> ≤2023|111|83.8
>
> Stable performance on 2024-2025 cases and SOTA results on our **OOD test set** (Table 10) proving that Salus learns the *strategic reasoning process*, rather than merely memorization.
>
> ---
> ### **W5: Human Verification & Teacher-Guided Rollouts**
> **1. Expert Details:** The test set was verified by 15 human physicians (average experience: >10 years). We used a cascade filtering process:
> Source|Before LLM Filter|After LLM but Before Human Filter|Final Retained
> -|-|-|-
> NEJM|131|128|123
> JTO|290|70|62
> JAMA|124|86|84
> iiYi|521|521|483
>
> The agreement rate of experts is 93.4%. And LLM filters (DeepSeek-V3.1 & GPT-5 mini) had a Fair agreement (Cohen’s Kappa = 0.25).
>
> **2. Why Teacher-Guided Rollouts?** Manual trajectories for 8,645 training cases are cost-prohibitive. We use the teacher model for *scalable cold-start SFT*. Our RL phase, guided by the human-aligned LLM Judge, corrected teacher biases, allowing Salus-7B to eventually outperform its teacher DeepSeek-V3.1.
>
> ---
> ### **W6 & W7: Test Cost and Clinical Relevance**
> We integrated a **diagnostic cost ($)** metric into our evaluation:
> Model|Top-1 (%)|Inf. Suff. (%)|#Rounds|#Tests|**Price ($)**
> -|-|-|-|-|-
> GPT-5.2|80.3|71.0|1.8|27.6|363
> Baichuan-M2-32B|48.7|56.5|3.1|26.0|300
> **Salus-7B (Ours)**|**83.6**|**72.9**|2.9|20.0|**282**
> DeepSeek-V3.2|71.4|66.9|1.9|12.0|156
>
> **Analysis:**
> 1. **Efficiency:** Salus-7B achieves the highest accuracy with *fewer* tests and *lower* costs than GPT-5.2 and Baichuan-M2-32B.
> 2. **Clinical Relevance (W7):** We do not use the original case report tests as a rigid "Gold Standard" because case reports often omit **negative findings**, which are clinically crucial for ruling out differential mimics. Therefore, Salus's combination of high *Information Sufficiency (72.9%)* and a *controlled test count/cost* is the evidence that its requested tests are highly relevant and discriminant, effectively balancing diagnostic safety and efficiency.
>
> **References:**
>
> [1] Zhu et al. "DiagnosisArena: benchmarking diagnostic reasoning for large language models."  arXiv:2505.14107 (2025).
>
> [2] Qiu et al. "Evolving Diagnostic Agents in a Virtual Clinical Environment."  arXiv:2510.24654 (2025).
>
> [3] Nori et al. "Sequential diagnosis with language models."  arXiv:2506.22405 (2025).
>
> [4] Schmidgall et al. "Agentclinic: a multimodal agent benchmark to evaluate ai in simulated clinical environments."  arXiv:2405.07960 (2024).
>
> We sincerely hope these new experiments and clarifications address your concerns.
>
> Best regards,
>
> Authors of Submission 23861

---

> > ### Author Rebuttal · Reviewer_6Qi3 · 2026-04-04
> >
> > I thank the authors for the rebuttal. While the results look nice, the consistent reliance on LLMs in generating medical trajectories, judging, etc. without checks can be dangerous. I maintain my score and still lean towards Weak Accept.

---

### Official Review · Reviewer_TNw4 · 2026-03-13

**Soundness:** 3
**Presentation:** 3
**Significance:** 3
**Originality:** 3
**Overall Recommendation:** 4
**Confidence:** 4

**Summary:**

The paper introduces CompDiag-Bench, a benchmark where a model must strategically request diagnostic tests from a dynamic environment in order to reach a definitive diagnosis. To address this setting, the authors propose Salus, a multi-agent framework that decomposes diagnostic reasoning into three roles: a Differential Reasoner, a Strategic Controller, and a Workup Proposer. The system is trained using multi-agent reinforcement learning with structured rewards, leveraging an LLM-as-a-Judge to provide dense feedback that penalizes premature diagnostic closure and encourages accurate differential diagnosis. Empirically, the approach shows performance comparable to GPT-5.2 while using a 7B-scale model.

**Compliance With Llm Reviewing Policy:**

Affirmed.

**Final Justification:**

Most of my concerns were adequately addressed in the rebuttal, and I therefore maintain my positive score.

**Key Questions For Authors:**

1) It appears that the LLM does not maintain a persistent conversational context across agent interactions. Instead, each role (DR, SC, WP) receives a newly constructed prompt containing the current clinical state and optional outputs. Does this mean that the interaction history is reconstructed from the current state H_t at each step, rather than preserved as a full conversational trajectory?

2) If yes for 1), how does the approach mitigate potential information loss from trajectory decomposition, particularly with respect to long-range dependencies in diagnostic reasoning?

3) Is training / using one single model with different instructions considered as truly multi-agent system? How would the performance differ if trained for / used separate weights?

If the authors can clarify these points, I believe the paper would be significantly strengthened and I would consider increasing my score.

**Limitations:**

The evaluation lacks multi-agent baselines and the step-level training formulation may overlook trajectory-level dependencies that are important for long-horizon diagnostic reasoning.

**Strengths And Weaknesses:**

Strengths:

The motivation is clear in both the high-level framework and the low-level details and implementations (e.g. high linguistic variability in medical literature) and the paper is well written. Evaluating AI systems in interactive diagnostic settings is an important direction that remains relatively underexplored.

The paper contributes both a benchmark and a multi-agent RL framework. The experimental results showing competitive performance with a 7B model are notable. The RL formulation also appears thoughtfully designed given the high cost of multi-agent interaction during training.

The inclusion of SFT ablation studies is useful for understanding the contribution of different training components and may be helpful for researchers implementing RL-based agent systems.

Weaknesses:

The evaluation compares against single LLM baselines, while other multi-agent systems are not considered in baselines. This makes it difficult to isolate the benefit of the specific Salus design.

The training formulation appears to operate at the step level rather than the full trajectory level, which may limit the model’s ability to capture long-horizon dependencies in diagnostic reasoning.

---

> ### Author Rebuttal · Authors · 2026-03-31
>
> Dear Reviewer TNw4,
>
> We are grateful for your constructive feedback, which is helpful in improving our work. Based on your suggestions, we have conducted targeted new experiments and detailed analyses that address your concerns and provide a more comprehensive validation of the Salus framework.
>
> ---
> ### **W1: Comparison with Multi-Agent Baselines**
> As you insightfully pointed out, evaluating against other multi-agent systems is crucial to isolate the benefits of Salus's specific design. We compared Salus-7B against 3 recent multi-agent baselines implemented using **DeepSeek-V3.2** as their backbone LLM: MedAgents[1], MDAgents[2], and DeepRare[3].
>
> **Context & Experimental Setup:** Existing medical multi-agent systems are primarily designed for *single-round, static diagnostic tasks* and lack the mechanisms such as our Strategic Controller and Workup Proposer to navigate multi-round dynamic testing. We therefore compared methods under a static diagnosis setting on 269 high-complexity cases, where agents receive the full clinical record and predict the final diagnosis.
>
>
> **Results:** Our findings clearly illustrate the superiority of our multi-agent design.
> Framework|Paradigm|Top-1(%)|Top-2(%)|Top-3(%)
> -|-|-|-|-
> MedAgents|Multi-expert Collaboration|71.7|79.6|84.0
> MDAgents|Adaptive Collaboration|71.7|83.6|86.6
> DeepRare|Autonomous Rare Disease Team|78.4|82.5|85.5
> **Salus-7B(Ours)**|**DR-SC-WP**|**86.6**|**90.7**|**94.1**
>
> **Analysis:** Even when powered by a frontier model (DeepSeek-V3.2), 3 baselines fall short of Salus-7B. These results suggest that existing multi-agent designs rely more on role specialization than on strengthening reasoning, whereas Salus improves reasoning more effectively through RL.
>
> ---
> ### **Q1, Q2 & W2: Context Maintenance and Long-Horizon Dependencies**
> Thank you for raising this critical question. We would like to clarify that **$H_t$ does not discard the conversational trajectory; rather, it is the exact, cumulative interaction history.**
>
> **1. Mechanism of $H_t$:** During the interaction, $H_t$ explicitly records the step-by-step sequential trajectory. Its format is strictly maintained as follows:
> ```
> Initial Clinical Context:** [Patient History, Physical Exam...]
> Round 1:
> Doctor: Request tests: [A, B, C]
> Environment: Test results of [A, B, C] are...
> Round 2:
> Doctor: Request tests: [D, E]
> Environment: Test results of [D, E] are...
> ...
> ```
>
> Therefore, because every exact test result is chronologically appended to $H_t$, **there is absolutely no clinical information loss** across the trajectory. The long-horizon dependencies are preserved in the text.
>
> **2. Robustness in Training vs. Evaluation:** To further ensure the model does not suffer from "lost-in-the-middle" issues over long horizons, we explicitly designed the training phase to expose the backbone to diverse formats of $H_t$ including raw interaction history, structured patient records, and case summaries. During evaluation, we strictly use the chronological interaction history to guarantee zero information loss. This approach effectively translates the long-horizon Markov Decision Process into a sufficient Markov state at each step, ensuring stable RL optimization without sacrificing trajectory-level dependencies.
>
> ---
> ### **Q3: Shared Weights vs. Separate Weights in the Multi-Agent System**
>
> To assess the effect of separate weights, we independently trained three separate 7B models (`DR-7B` via SFT+RL, `SC-7B` via SFT+RL, and `WP-7B` via SFT) and deployed them collaboratively.
>
> Framework|Weight Strategy|Top-1(%)|Inf. Suff.(%)
> -|-|-|-
> Separate Weights|3 Independent 7B Models|63.6|67.3
> **Salus-7B (Ours)**|**Shared Weights (1 Model)**|**83.6**|**72.9**
>
> **Analysis:**
> 1. **Positive Transfer of Medical Knowledge:** The results show that sharing weights significantly outperforms using separate weights. This is because the tasks of DR (reasoning), SC (termination control), and WP (testing) are intrinsically connected through the same underlying distribution of medical knowledge. When the backbone's medical reasoning capability is enhanced via gradients from the DR task, the SC and WP agents simultaneously benefit from this deepened medical understanding. Separate training isolates these gradients, preventing this crucial **positive transfer**.
> 2. **Resource Efficiency:** Furthermore, separate weights also require roughly 3× training and inference resources.
>
> **References**
>
> [1] Tang et al. MedAgents: Large language models as collaborators for zero-shot medical reasoning. ACL 2024
>
> [2] Kim et al. MDAgents: An adaptive collaboration of llms for medical decision-making. NeurIPS 2024
>
> [3] Zhao et al. An agentic system for rare disease diagnosis with traceable reasoning. Nature 2026
>
> ---
> Once again, we sincerely thank you for your detailed review. We hope that our comprehensive responses and new experiments have addressed your concerns and further clarified the significance of our work.
>
> Best regards,
>
> Authors of Submission 23861

---

> > ### Author Rebuttal · Reviewer_TNw4 · 2026-04-03
> >
> > Regarding W1,
> > -> The presented experimental results appear to strengthen the method, and it would be beneficial to include comparisons against agentic system baselines in the manuscript.
> >
> > Regarding Q1, Q2, and W2,
> > -> Does not the manuscript state that previous history is discarded during training to reduce cost? I would appreciate a more detailed explanation of how history is handled during both training and inference, as well as across all agents throughout the interaction, including the inputs and outputs of each agent at training and test time.
> >
> > Regarding Q3,
> > -> The presented experimental results seem to address the question of single model vs separate models.

---

> > > ### Author Response · Authors · 2026-04-06
> > >
> > > Dear Reviewer TNw4,
> > >
> > > Thank you for your constructive feedback. Next we provide a detailed explanation of the handling of clinical history ($H_t$).
> > >
> > > ---
> > > ### **1. Rationale for Not Compressing $H_t$**
> > > We would like to clarify a potential misunderstanding: our strategy for cost reduction involves **discarding the intermediate Chain-of-Thought (CoT) reasoning from previous rounds**, rather than the **clinical evidence ($H_t$) itself**. The reasons for maintaining the full history $H_t$ instead of further compressing it are as follows:
> > > *   **Token Pricing:** For modern LLMs (e.g., DeepSeek-V3.1), output tokens are significantly more expensive than input tokens, often by a factor of 3:1 or more. Since the primary computational cost in our system is dominated by the generation of long-form CoT reasoning, the savings gained from compressing input tokens ($H_t$) are marginal.
> > > *   **Minimal Compression Gains:** Summarizing multi-round dialogues into paragraphs typically yields only a ~20% reduction in token count, which poses the potential risk of losing subtle but critical clinical nuances.
> > > *   **Selective Discarding:** Instead of summarization, we optimize input token cost by **removing previous reasoning (CoT)** while retaining all **objective findings and actions**. This ensures the input remains factual and manageable without information loss.
> > >
> > > ### **2. Input/Output (I/O) Flow of Multi-Agent Interaction**
> > > Each agent in our system utilizes the $H_t$ as its core context. The detailed prompt templates are available in **Appendix F**. The I/O flow for each round $t$ is summarized below:
> > > Agent|Input|Output
> > > -|-|-
> > > **DR**|Instruction + $H_t$|`<think>` CoT `</think>` + Differential Diagnoses List $D_t$
> > > **SC**|Instruction + $H_t$ + $D_t$|`<think>` CoT `</think>` + Signal $s_t$ (Stop/Continue)
> > > **WP**|Instruction + $H_t$ + $D_t$|`<think>` CoT `</think>` + Test Request List $E_t$
> > >
> > > Upon receiving test results of $E_t$ from the Environment, $H_t$ is updated via concatenation:
> > > $$ H_{t+1} \leftarrow H_{t} \oplus (E_t, \text{Result}_t) $$
> > >
> > > **Here is an Illustrative Example of $H_t$ Growth ($H_2 \rightarrow H_3$).**
> > >
> > > **$H_2$:**
> > > >**Initial Clinical Context:**
> > > >45-year-old male, chronic cough ×3 months, hemoptysis ×1 week, 30 pack-year smoking history. Normal O2 sat.
> > > >
> > > >**Round 1:**
> > > >Doctor: Request tests: [CXR, CBC]
> > > >
> > > >Environment: CXR: Left lung mass; CBC: Within Normal Limits.
> > > >
> > > >**Round 2:**
> > > >Doctor: Request tests: [CT Chest]
> > > >
> > > >Environment: CT: 3cm spiculate lesion, hilar lymphadenopathy.
> > >
> > > If the WP agent requests: [CT-guided Biopsy], then $H_t$ is updated:
> > >
> > > **$H_3$:**
> > > >**Initial Clinical Context:**
> > > >45-year-old male, chronic cough ×3 months, hemoptysis ×1 week, 30 pack-year smoking history. Normal O2 sat.
> > > >
> > > >**Round 1:**
> > > >Doctor: Request tests: [CXR, CBC]
> > > >
> > > >Environment: CXR: Left lung mass; CBC: Within Normal Limits.
> > > >
> > > >**Round 2:**
> > > >Doctor: Request tests: [CT Chest]
> > > >
> > > >Environment: CT: 3cm spiculate lesion, hilar lymphadenopathy.
> > > >
> > > >**Round 3:**
> > > >Doctor: Request tests: [CT-guided Biopsy]
> > > >
> > > >Environment: Pathology: Adenocarcinoma.
> > >
> > >
> > > ### **3. Representation and Handling of $H_t$: Training vs. Testing**
> > > To ensure both diagnostic rigor and clinical flexibility, we designed $H_t$ to support three distinct formats. While the **Multi-round Dialogue** serves as the default format, we use **Qwen2.5-7B-Instruct** to transform the dialogues into **Summaries** (mimicking medical abstracts) and **Structured Records** (Electronic Health Record, EHR format including Chief Complaint, History of Present Illness, Physical Exam Results, and Lab Results).
> > >
> > > The utilization of these formats differs between stages:
> > > *   **During Training:** We randomly sample $H_t$ from all three formats. This "multi-format training" strategy ensures the model's **input robustness**, allowing it to adapt to various real-world clinical entry points. For instance, a human clinician can provide an existing EHR report or a brief narrative summary, and our system can immediately initiate reasoning without being restricted to a specific dialogue structure.
> > > *   **During Testing:** To ensure zero information loss and fair comparison, we strictly utilize the **Multi-round Dialogue** format. At each round, the latest test results provided by the environment are directly appended to the history $H_t$, maintaining an exhaustive and chronological record of the diagnostic trajectory.
> > >
> > > ---
> > >
> > > We hope these details resolve your concerns.
> > >
> > > Best regards,
> > >
> > > Submission 23861 Authors

---

### Official Review · Reviewer_PWxF · 2026-03-13

**Soundness:** 3
**Presentation:** 2
**Significance:** 3
**Originality:** 3
**Overall Recommendation:** 4
**Confidence:** 3

**Summary:**

Paper studies complex medical diagnosis as a sequential test-selection problem rather than a static question-answering task. It introduces CompDiag-Bench, an interactive benchmark built from curated clinical records in which an AI clinician iteratively requests tests from an LLM-based diagnostic environment, and proposes Salus, a three-role framework with a Differential Reasoner, Strategic Controller, and Workup Proposer trained with teacher-generated supervised data plus step-wise GRPO using LLM-judge rewards. Experiments on high-complexity, routine, and out-of-distribution subsets compare Salus-7B to open and proprietary baselines and report large gains over the untuned backbone and SFT-only variants.

**Compliance With Llm Reviewing Policy:**

Affirmed.

**Final Justification:**

Thanks for authors' rebuttal, my concerns were mostly addressed, and i have updated ratings accordingly.

**Key Questions For Authors:**

What agreement do human clinicians have with the LLM judge on the four-level diagnosis rubric and with the oracle-based information-sufficiency labels, at least on a stratified sample of hard cases?

Since RL samples states from SFT data rather than model-induced multi-round trajectories, what evidence shows that the learned policy works well on its own state distribution, such as a state-mismatch analysis or a small on-policy or full-trajectory RL comparison?

Can you provide a same-backbone single-agent baseline trained with the same SFT data and reward sources, and justify or ablate the choice to keep the Workup Proposer frozen during RL?

**Limitations:**

paper already discusses text-only scope, cost, environment fidelity, hallucination, and bias, but it omits several material limitations. Most importantly, it should explicitly acknowledge that undocumented tests receive synthetic results constrained to be consistent with the gold diagnosis, which may favor models that order more tests and inflate both accuracy and information-sufficiency metrics. It should also discuss the lack of human validation for the LLM judge and oracle pipeline, as well as the limited contamination analysis for public-source cases. A stronger limitations section would report the fraction of synthetic tests, add record-only or ambiguous-result ablations, include a clinician agreement study, and clarify decontamination checks and API-version dependencies.

**Strengths And Weaknesses:**

Strengths:
- targets a clinically meaningful gap in current evaluation by modeling diagnosis as sequential evidence gathering with an explicit stop-or-continue decision, which directly captures premature closure and makes the task more realistic than static QA benchmarks.
- Salus decomposition is clearly specified through formal state definitions, role descriptions, prompts, and reward functions, which makes the method easier to audit and helps readers understand what each module is expected to contribute.


Weaknesses:

evaluation pipeline depends heavily on LLM-generated infrastructure: DeepSeek-V3.1 serves as environment, judge, and oracle in the main setup, and DeepSeek-V3.2 provides RL rewards. Cross-model robustness with Gemini is helpful, but without human agreement studies on the diagnosis rubric and information-sufficiency labels, it remains unclear how stable the reported gains are with respect to evaluator preferences and borderline cases.

methodological claim is somewhat broader than the evidence shown. RL samples states from the SFT datasets and optimizes step-wise DR and SC outputs, rather than training on full model-induced multi-round trajectories with environment interaction, so the paper more clearly demonstrates effective offline post-training on teacher-derived states than a solution to long-horizon exploration and credit assignment; the missing evidence is either an on-policy or full-trajectory RL comparison, or direct analysis of train-test state-distribution mismatch.

significance is limited by the current efficiency story. Salus often uses more rounds and many more tests than leading baselines, and the paper does not provide a budgeted evaluation or accuracy-cost Pareto analysis, so it is difficult to judge whether the learned strategy remains attractive once testing cost, delay, or invasiveness are constrained.

---

> ### Author Rebuttal · Authors · 2026-03-31
>
> Dear Reviewer PWxF,
>
> We are grateful for your constructive feedback. We have conducted new experiments to address your concerns and validate our multi-agent RL framework.
>
> ---
> ### **W1 & Q1: Human-LLM Agreement**
> We conducted a blind study with 3 board-certified physicians (Respiratory, Gastroenterology, Cardiology) evaluating 240 diagnoses sampled from the test trajectories using our 4-level rubric. Cohen's Kappa between the LLM judge and human physicians is calculated.
> - **4-level Agreement:** Cohen's Kappa = **0.64 (Substantial Agreement)**.
> - **Binary Agreement (Correct + Clinically Acceptable vs. Partially Correct + Incorrect):** Cohen's Kappa = **0.81 (Almost Perfect Agreement)**.
> This confirms that our Diagnostic Accuracy metric is highly aligned with expert clinical judgment.
>
> **Information Sufficiency:** We clarify that Information Sufficiency is a *relative indicator* of how completely clinical evidence was gathered. Different Oracle models have different diagnostic capacities; thus, this metric cannot be directly compared to human doctors.
>
> ---
> ### **W2 & Q2: State Distribution and RL Methodology**
> **1. State Mismatch Analysis:** We analyzed the cumulative history states ($H_t$) in Training vs. Testing.
> Dataset|Metric|Mean|Std|Median|Min|Max|Coverage Rate
> -|-|-|-|-|-|-|-
> Train|#Tests|8.6|9.5|6|0|63|-
> Test|#Tests|11.0|11.7|8|0|60|**100%**
> Train|#Tokens|1628.6|540.8|1540|429|4724|-
> Test|#Tokens|1460.4|808.2|1284|279|5543|**96.6%**
>
> The test distribution is largely covered by the training distribution (Coverage > 96%) in terms of both the number of requested tests and token counts, indicating no severe state mismatch.
>
> **2. Why not full-trajectory RL?** Full-trajectory RL requires serial execution: WP proposes tests $\rightarrow$ wait for Environment API generation $\rightarrow$ DR/SC acts, which introduces massive latency and leaves GPUs idle. The full-trajectory RL for one epoch would take **> 14 days**; however, our step-wise RL takes **<2 days** while maintaining SOTA performance.
>
> ---
> ### **Q3 (Part 1): Same-Backbone Single-Agent Baseline**
> We trained a single-agent baseline (Qwen2.5-7B-Instruct) through SFT + RL (using the same data and DR rewards).
> Model|Top-1 (%)|Inf. Suff. (%)|#Rounds
> -|-|-|-
> **Salus-7B (Multi-Agent SFT + RL)**|**83.6**|**72.9**|2.9
> Salus-Zero-7B (Multi-Agent SFT)|55.0|65.8|2.5
> Single-Agent (SFT + RL)|53.5|67.1|3.0
> Single-Agent (SFT)|53.5|63.6|3.2
>
> **Analysis:** **RL brings a slight improvement over Single-Agent** due to *reward sparsity*—it only receives a reward at the final turn of a long trajectory. Our multi-agent architecture enables dense, step-wise rewards.
>
> ---
> ### **Q3 (Part 2): Justification for Freezing the Workup Proposer (WP)**
> Designing rewards for the WP is challenging due to reward hacking. In our early explorations, when we applied RL to the WP, we observed severe hacking behaviors:
> 1. *Bundling:* bundling multiple tests into a single test to decrease number of tests.
> 2. *Repetition:* requesting tests already present in the history to steal "safe" rewards.
>
> We restricted the above actions and designed reward: *Reward = number of differential diagnoses ruled out by the requested tests.* However, as shown below, applying RL to the WP degraded diagnostic accuracy:
> Model Variant|Top-1|Top-2|Top-3|Inf. Suff.|#Rounds
> -|-|-|-|-|-
> **Salus-7B (Frozen WP)**|**83.6**|**86.6**|**88.9**|**72.9**|2.9
> Salus-7B + RL for WP|80.1|84.4|86.3|72.1|2.7
>
> Thus, freezing WP is a deliberate and empirically supported choice.
>
> ---
> ### **W3: Cost Analysis**
> We calculate the total price of requested test items.
> Model|Top-1 (%)|Inf. Suff. (%)|#Rounds|#Tests|**Price ($)**
> -|-|-|-|-|-
> GPT-5.2|80.3|71.0|1.8|27.6|363
> Baichuan-M2-32B|48.7|56.5|3.1|26.0|300
> **Salus-7B (Ours)**|**83.6**|**72.9**|2.9|20.0|**282**
> DeepSeek-V3.2|71.4|66.9|1.9|12.0|156
>
> **Salus-7B is more cost-effective than GPT-5.2 and Baichuan-M2**, while achieving higher accuracy. Salus-7B balances accuracy and cost efficiently, proving it does not rely on infinite budgets.
>
> ---
> ### **Limitations: Synthetic Results vs. Record-Only Environment**
> In our benchmark, **50%~80% of test results are synthetic**.
>
> We avoid "Record-Only" (returning "Not Available") to prevent Negative Information Leakage: clinical records typically only document *positive/abnormal* findings, so "Not Available" implicitly signals a test is irrelevant, leaking the diagnosis path [1].
>
> We also conducted a human audit of 100 randomly sampled results from the Environment. **Only 7 cases contained potentially diagnostic terms**, and all 7 cases were pathology/biopsy reports (e.g., returning "malignant cells" for a cancer patient), which actually aligns with real-world clinical feedback.
>
> **Reference**
>
> [1] Nori et al., "Sequential diagnosis with language models." arXiv:2506.22405 (2025).
>
> ---
> Once again, we sincerely thank you for your feedback. We hope these responses and new data address your concerns.
>
> Best regards,
>
> Authors of Submission 23861

---

> > ### Author Rebuttal · Reviewer_PWxF · 2026-04-03
> >
> > Thanks for authors' rebuttal, my concerns were mostly addressed, and i have updated ratings accordingly.

---

### Decision · Program_Chairs · 2026-04-30

**Decision:**

Accept (regular)

**Comment:**

The paper reframes complex medical diagnosis as a sequential test-selection process rather than a static question-answering task. It introduces CompDiag-Bench, an interactive benchmark derived from curated clinical records, where an AI clinician iteratively requests diagnostic tests within an LLM-based environment. The authors propose Salus, a three-component framework comprising a Differential Reasoner, Strategic Controller, and Workup Proposer, trained using teacher-generated supervised data alongside step-wise GRPO with LLM-judge rewards. In the rebuttal phase, the majority of reviewer questions have been addressed and reviewers are happy with the author responses overall. My key concern for this paper is where the key technical challenge lies to warrant acceptance to ICML.